# Simple sequence repeats drive genome plasticity and promote adaptive evolution in penaeid shrimp

Jianbo Yuan[1,2,3,9], Xiaojun Zhang [1,2,3,9], Min Wang[4,5,9], Yamin Sun[6], Chengzhang Liu[1,2,3], Shihao Li[1,2,3], Yang Yu[1,2,3], Yi Gao[1,2,3], Fei Liu[1,2,3], Xiaoxi Zhang[1], Jie Kong[7], Guangyi Fan[8], Chengsong Zhang[1,2,3], Lu Feng [4,5✉], Jianhai Xiang [1,2,3✉] & Fuhua Li [1,2,3✉]

Simple sequence repeats (SSRs) are rare (approximately 1%) in most genomes and are generally considered to have no function. However, penaeid shrimp genomes have a high proportion of SSRs (>23%), raising the question of whether these SSRs play important functional and evolutionary roles in these SSR-rich species. Here, we show that SSRs drive genome plasticity and adaptive evolution in two penaeid shrimp species, *Fenneropenaeus chinensis* and *Litopenaeus vannamei*. Assembly and comparison of genomes of these two shrimp species at the chromosome-level revealed that transposable elements serve as carriers for SSR expansion, which is still occurring. The remarkable genome plasticity identified herein might have been shaped by significant SSR expansions. SSRs were also found to regulate gene expression by multi-omics analyses, and be responsible for driving adaptive evolution, such as the variable osmoregulatory capacities of these shrimp under low-salinity stress. These data provide strong evidence that SSRs are an important driver of the adaptive evolution in penaeid shrimp.

[1] CAS Key Laboratory of Experimental Marine Biology, Institute of Oceanology, Chinese Academy of Sciences, Qingdao, China. [2] Laboratory for Marine Biology and Biotechnology, Qingdao National Laboratory for Marine Science and Technology, Qingdao, China. [3] Center for Ocean Mega-Science, Chinese Academy of Sciences, Qingdao, China. [4] The Key Laboratory of Molecular Microbiology and Technology, Ministry of Education, Tianjin, China. [5] TEDA Institute of Biological Sciences and Biotechnology, Nankai University, TEDA, Tianjin, China. [6] Tianjin Biochip Corporation, Tianjin 300457, China. [7] Key Laboratory for Sustainable Utilization of Marine Fisheries Resources of Ministry of Agriculture, Yellow Sea Fisheries Research Institute, Chinese Academy of Fishery Sciences, Qingdao, China. [8] BGI-Qingdao, BGI-Shenzhen, Qingdao, China. [9] These authors contributed equally: Jianbo Yuan, Xiaojun Zhang, Min Wang. ✉email: fenglu63@nankai.edu.cn; jhxiang@qdio.ac.cn; fhli@qdio.ac.cn

Simple sequence repeats (SSRs) are repetitive sequences composed of tandem repetitions of short motifs (1–6 bp). SSRs account for only ~1% of the genome in most sequenced species[1,2], and have been commonly regarded as non-coding DNA, since they do not contain functional genomic information[3]. Recently, the genomes of many species have been published, revealing multiple species with genomes containing large proportions of SSRs[2,4,5], e.g., the body louse *Pediculus humanus* (10.52%) and the Californian leech *Helobdella robusta* (6.36%). The genome of a penaeid shrimp *Litopenaeus vannamei* is particularly notable for having the highest proportion of SSRs (~23.93%) among sequenced animal genomes up to now. Furthermore, a high proportion of SSRs (~10%) have been identified in the draft genomes of other penaeid shrimp[6]. Thus, these findings raise questions about the functions of these significantly expanded SSRs, including their role in the adaptation and evolution of SSR-rich species. In addition, it is unclear how these SSRs originated and expanded.

One of the most widely accepted hypotheses for the origin of SSRs is the DNA polymerase slippage model[7]. During DNA replication or repair, DNA polymerase slippage can occur when one DNA strand temporarily dissociates from the other, which leads to an increase in the number of repeats[1]. Slippage mutations have been demonstrated in many species but appear to lead to only small changes in motif repeat number[8]. A few SSRs could reach 30 motif repeats because of the existence of the mismatch repair (MMR) system[9–11]. Even escape from the MMR system, slippage mutations are unlikely to cause a major expansion of SSRs throughout the genome of SSR-rich species[12]. Thus, aside from slippage mutations, additional mechanisms underlying SSR expansion may exist in SSR-rich species.

Generally, SSRs were considered to have no functions in the genome, whereas many studies support an evolutionary role for some SSRs as important sources of adaptive genetic variation[3]. These SSRs might provide an evolutionary advantage for fast adaptation to new environments by serving as evolutionary "tuning knobs"[3,13]. These SSRs with putative functions might include regulation of genomic structures and gene expression[14]. Indeed, SSR variation has been shown to correlate with variations in social behavior phenotypes, skeletal morphology, and adaptive divergence of different populations[15–17]. The potential positive effects of SSRs on adaptive evolution of organisms are particularly interesting in SSR-rich species. The penaeid shrimp, whose Penaeidae ancestor originated in the Late Devonian[18,19], are especially interesting in this regard as this group might have undergone at least three worldwide mass extinctions during its evolutionary history[2,20]. In addition, these species present different ecological distributions and environmental adaptability traits[21]. Thus, the genomes of penaeid shrimp might provide us with an important opportunity to investigate whether SSR expansion contributed to the adaptive evolution and divergence of penaeid shrimp species.

Genomic plasticity, including genome copy number variation, inter- and intrachromosomal rearrangements, DNA insertions and deletions, and loss of heterozygosity, is well known in eukaryotes[22]. In this study, we aimed to determine whether the significant expansion of SSRs has driven genome plasticity and adaptive evolution in penaeid shrimp genomes. Here, based on the genome described in our previous study[2], we updated the genome assembly of *L. vannamei* to the chromosome level and de novo assembled the genome of another important penaeid shrimp, *Fenneropenaeus chinensis*. Comparative genomic analyses were performed to thoroughly investigate the functional and evolutionary roles of the SSRs of these two genomes, including functions in genome rearrangement and environmental adaptation. The expansion of SSRs drives the rapid adaptive evolution

**Table 1 Summary of two penaeid shrimp genome assemblies and annotations.**

| | *F. chinensis* | *L. vannamei* |
|---|---|---|
| **Genome assembly** | | |
| Chromosome number | 88 (2*n*) | 88 (2*n*) |
| Total length | 1,581,129,620 bp | 1,631,536,563 bp |
| Scaffold number | 8768 | 28,409 |
| Contig N50 | 58,996 bp | 57,650 bp |
| Scaffold N50 | 28,916,617 bp | 31,296,514 bp |
| GC content | 36.45% | 35.68% |
| **Genome annotation** | | |
| Transposable elements | 19.12% | 16.17% |
| SSRs | 19.50% | 23.93% |
| Protein-coding genes | 26,343 | 25,596 |

and divergence in the ancestor of penaeid shrimp, and our multiomic analyses revealed that some SSRs have important functions in response to environmental stress.

## Results

**Significant SSR expansion is a common characteristic in penaeid shrimp species.** To explore whether the SSR expansion originated from the ancestor of penaeid shrimp, we performed genome assembly and comparative analyses in two penaeid species (Fig. 1a). The *F. chinensis* genome size was estimated to be 1.88 Gb (Supplementary Figs. 1 and 2). We applied multiple sequencing technologies following the method described in our previous study[2] to overcome the extreme difficulties in genome assembly of penaeid shrimp species. Based on the various sequencing data (Supplementary Tables 1–3), a high-quality de novo reference genome assembly of *F. chinensis* was obtained with a total length of 1.58 Gb and a contig N50 of 59.00 kb, comparable to that of the previously constructed genome assembly of *L. vannamei*[2]. The genome assembly of *F. chinensis* was further refined using Hi-C data to comprise 8768 scaffolds with a scaffold N50 of 28.91 Mb (Table 1 and Supplementary Table 4). The assembly showed high integrity and quality as assessed by the Illumina read coverage (91.12%), unigenes from transcriptomes (94.83%), and conserved core arthropod genes (92.68%) (Supplementary Tables 5–7).

To perform comparative genomics analyses at the chromosome-level, we also refined the genome assembly of *L. vannamei* using Hi-C data (196×), and the total length of the final assembly was 1.63 Gb with a scaffold N50 of 31.30 Mb (Supplementary Fig. 3). To assess the accuracy of the assembly, we further compared the genome assembled based on the Hi-C data with a previously constructed high-density linkage map of *L. vannamei*[23]. As expected, high synteny was detected between the two chromosome-level genome assemblies (Fig. 1b and Supplementary Table 8).

Protein-coding gene prediction and annotation resulted in a total of 26,343 protein-coding genes annotated in the *F. chinensis* genome, with an average length of 7312 bp and 5.77 exons per gene; these measures were very similar to those of the *L. vannamei* genome (25,596 protein-coding genes, an average length of 8,889 bp, and 5.94 exons per gene; Supplementary Table 9 and Supplementary Fig. 4). Besides, the annotated genes in the two shrimp genomes showed very similar characteristics in gene structure (Supplementary Fig. 5) and functional classification distributions (Supplementary Figs. 6 and 7).

Repetitive sequences accounted for 48.58% in the genome of *F. chinensis*, and among them, transposable elements (TEs) accounted for 19.12%, which was close to that in the *L. vannamei* genome (16.17% of TEs; Supplementary Table 10 and

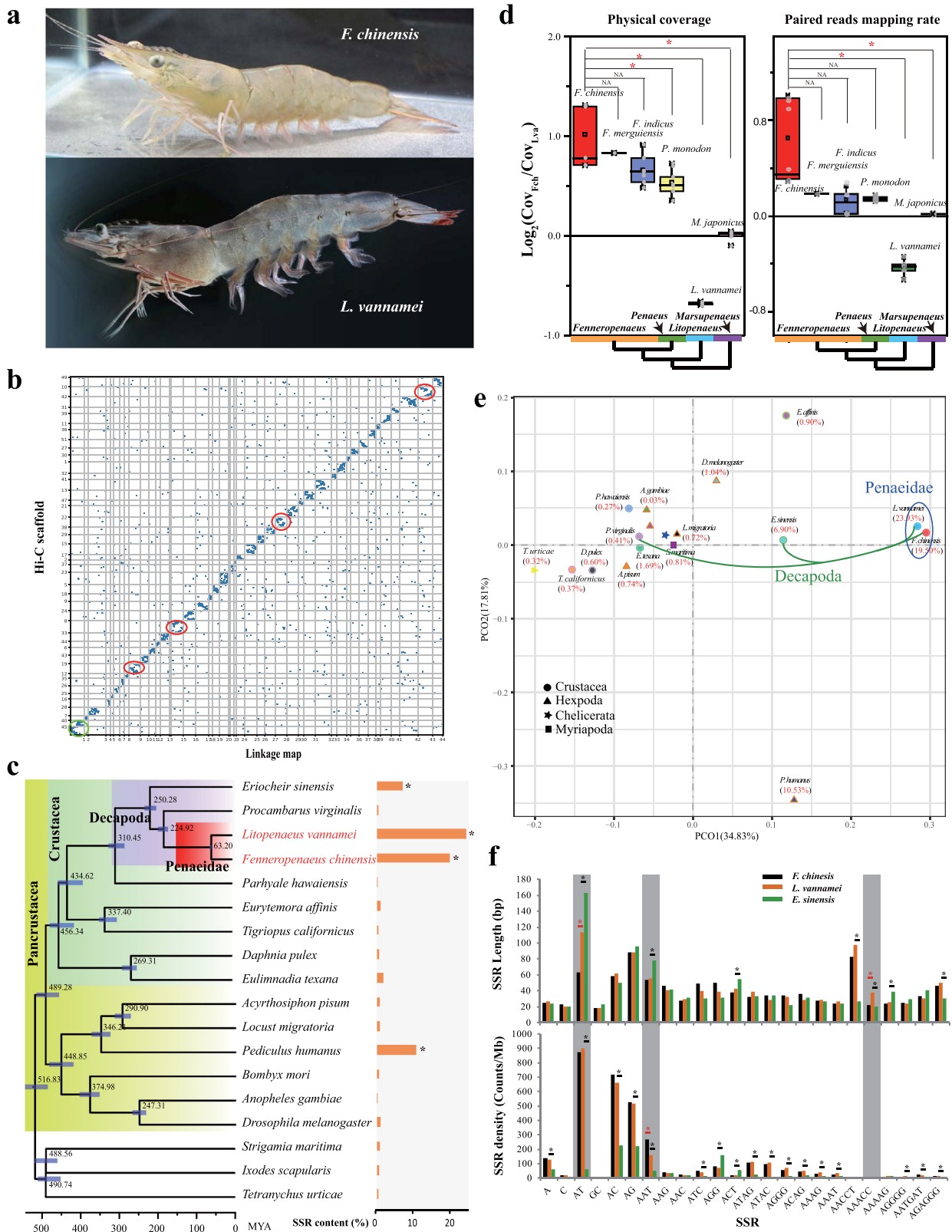

Supplementary Fig. 8). DNA transposons, especially En-Spm, in the two penaeid shrimp genomes were highly expanded compared to those in other decapods ($p < 0.01$). Comparison between the two genomes indicated that *F. chinensis* has a higher proportion of En-Spm elements (10.08%) than *L. vannamei* (6.39%, $p < 0.01$). In addition, more type I long interspersed nuclear elements (LINEs) (0.84%) and fewer hAT-Charlie elements (0.14%) were present in the *F. chinensis* genome than in the *L. vannamei* genome (LINE/I: 0.23% and hAT-Charlie: 1.00%; Supplementary Fig. 9). SSRs accounted for the highest percentage (19.50%) of the repetitive sequences in the *F. chinensis* genome, and the percentage was comparable to that in the *L. vannamei* genome (23.93%). The SSRs in these species were significantly higher than those in other decapod species ($p < 0.01$),

**Fig. 1 Comparative genomics of penaeid shrimp and other arthropods. a** The pictures of *F. chinensis* and *L. vannamei*. **b** Synteny of the chromosome-level genomes assembled based on Hi-C data and a high-density linkage map of *L. vannamei*. Rearrangements were identified in four pseudochromosomes (red circles), and a Hi-C scaffold was found to be composed of two linkage groups (green circle). **c** Phylogenetic placement of penaeid shrimp in the arthropod phylogenetic tree. The numbers on the branches indicate the estimated divergence times (MYA). Error bars indicate 95% confidence levels. The SSR content for each species is shown in the right bar plot. **d** Comparison of the physical coverage and read mapping rates of the two penaeid shrimp genomes based on the Illumina sequencing data of various penaeid shrimp species. A positive value of $\log_2(Cov_{Fch}/Cov_{Lva})$ indicates that the genome sequences of the corresponding species were more similar to that of *F. chinensis*, while a negative value indicates that the genome sequences of the corresponding species were more similar to that of *L. vannamei*. The line indicates the median value, the square symbol indicates the mean, the upper and lower box edges indicate the 75% and 25% quartiles, respectively, and the "X" indicates the outliers. * indicates significant difference between two groups of values ($p < 0.01$). **e** Principal component analysis (PCA) of SSR composition among arthropod genomes. **f** Comparison of SSR length and density of each type among three decapod genomes. The orange star indicates a significant variation ($p < 0.05$) between *F. chinensis* and *L. vannamei*, and the black star indicates a significant variation between *L. vannamei* and *E. sinensis*.

such as *Procambarus virginalis* (0.41%) and *Eriocheir sinensis* (6.90%), as well as in other published genomes (Fig. 1c and Supplementary Tables 11 and 12). These results suggest that significant SSR expansion is a common characteristic shared by penaeid shrimp species.

**Genome evolution with SSR variation in penaeid shrimp.** The penaeid shrimp are estimated to have diverged from other decapods approximately 225 million years ago (MYA) (Fig. 1c), after the Permian-Triassic mass extinction era (~252 MYA)[20], as evidenced by the fossils of the Luoping Biota[24]. Shrimp taxa, including the penaeid shrimp species *F. chinenesis* and *L. vannamei*, rapidly evolved in the Late Cretaceous after a mass extinction event[25]. Notably, *F. chinensis* and *L. vannamei* both belong to the same genus, *Penaeus*; six subgenera of *Penaeus* have been upgraded to the genus level in recent decades[26], but this change has not yet been completely adopted. Despite belonging to the same genus, *F. chinensis*, *L. vannamei*, and species in six subgenera displayed some diversity in morphology and genetic composition (Fig. 1d). The whole-genome sequencing reads showed significantly higher physical coverage and mapping rates within each subgenus (i.e. *Fenneropenaeus*) than between the subgenera (Supplementary Fig. 10 and Supplementary Table 13). To distinguish the species in different subgenera, it is better to upgrade their taxonomic levels. Thus, we used *Fenneropenaeus* and *Litopenaeus* as genus names in this study.

A large number of Penaeidae-specific (2726) and significantly expanded (36) gene families have been identified in the two penaeid shrimp genomes (Supplementary Fig. 11). These genes were enriched in functions related to the myosin and actin cytoskeleton, various organismal systems (e.g., endocrine and immune system), and some metabolic processes (e.g., metabolism of glycan, amino acid and lipid), and so on (Supplementary Data: Tables SS1 and SS2), reflecting the typical phenotypic changes that occur during adaptive evolution. They were expanded two times before and after the divergence of penaeid shrimps, respectively (Supplementary Fig. 12). Besides, 148 genes were identified under positive selection, which enriched in phototransduction and amino acid metabolism (Supplementary Fig. 13).

In addition to these expanded gene families, the significant expansion of SSRs was also an obvious characteristic shared by penaeid shrimp. Principal component analysis (PCA) indicated that the SSR structures of the two penaeid shrimp species were highly similar, and distinct from those of other 17 arthropods (Fig. 1e). When comparing with that of other SSR-rich species (Supplementary Table 11)[2], e.g. *P. humanus* (10.52% SSRs) and *H. robusta* (6.36% SSRs), we found that both the mean length (72.21 bp for *L. vannamei* and 56.54 bp for *F. chinensis*) and the density (3315.23/Mb for *L. vannamei* and 3449.10/Mb for *F. chinensis*) of SSRs were higher in the two penaeid shrimp

species, except that *P. humanus* had the highest SSR density at 4508.69/Mb (Supplementary Data: Tables SS3 and SS4). Moreover, the penaeid shrimp harbored many more dinucleotide SSRs $((AT)_n, (AC)_n, (AG)_n)$, while *P. humanus* had more A-rich SSRs $((A)_n, (AAT)_n, (AAAT)_n)$, and *H. robusta* had more triplet and tetranucleotide SSRs $((ATC)_n, (ATC)_n, (AAC)_n,$ and $(ATAC)_n)$ (Supplementary Fig. 14). Thus, the expansion of specific types of SSRs was an independent lineage-specific event rather than a convergent evolution event in SSR-rich species. Among the four available decapod genomes, *E. sinensis* also had a high SSR content (6.90%)[27], but the composition and structures of its SSRs (especially the SSR density) differed significantly from those of the penaeid shrimp species (Fig. 1f and Supplementary Table 12). Except for $(AGG)_n$ and $(ACT)_n$, the SSR density in the genome of *E. sinensis* was significantly lower than two shrimp species for most types of SSRs, especially dinucleotide SSRs. In contrast, the significant expansion of SSRs, especially dinucleotide SSRs, was similar in the two penaeid shrimp species, which suggested that these SSRs might stem from a common ancestor.

Even though the SSR compositions in these two shrimp species were quite similar, some variations were observed. The average length of $(AT)_n$ SSRs was longer in the *L. vannamei* genome (113.45 bp) than in the *F. chinensis* genome (62.95 bp), although their $(AT)_n$ densities were similar (Fig. 1f); thus, the overall $(AT)_n$ content in the genome of *L. vannamei* (10.21%) was twice of *F. chinensis* (5.51%; Supplementary Table 12). In contrast, *F. chinensis* had higher content and higher density of $(AAT)_n$ SSRs (1.45%, 267.44/Mb) than *L. vannamei* (0.88%, 158.33/Mb). Therefore, the $(AT)_n$ and $(AAT)_n$ are two major species-specific expanded SSRs in *L. vannamei* and *F. chinensis*, respectively. The major telomeric SSRs, $(AACCT)_n$, was more numerous and longer in *L. vannamei* than in *F. chinensis*[11], and a similar phenomenon was identified for another penta-nucleotide SSR, $(AAACC)_n$ (Fig. 1f). In total, 18,830 (71.48%) and 20,018 (78.28%) protein-coding genes containing SSRs were identified in *F. chinensis* and *L. vannamei*, respectively. Except for SSR-free genes, the SSR compositions in most of the orthologous genes of these two shrimp species are different.

Overall, since stem from the same genus, the two penaeid shrimp species shared similar genomic characteristics, including protein-coding genes and SSRs, which might inherit from their ancestor. However, some variations, especially species-specific expanded SSRs, were also identified between the two genomes, which occurred after their divergence.

**SSR expansion in penaeid shrimp genomes is associated with TE expansion.** To understand how SSRs expanded in penaeid shrimp, we analyzed the two shrimp genomes thoroughly. Generally, SSRs are short (<30 repeats) because of correction by MMR system[1,3]. However, the average length of dinucleotide SSRs (except $(CG)_n$) in the penaeid shrimp genomes was longer

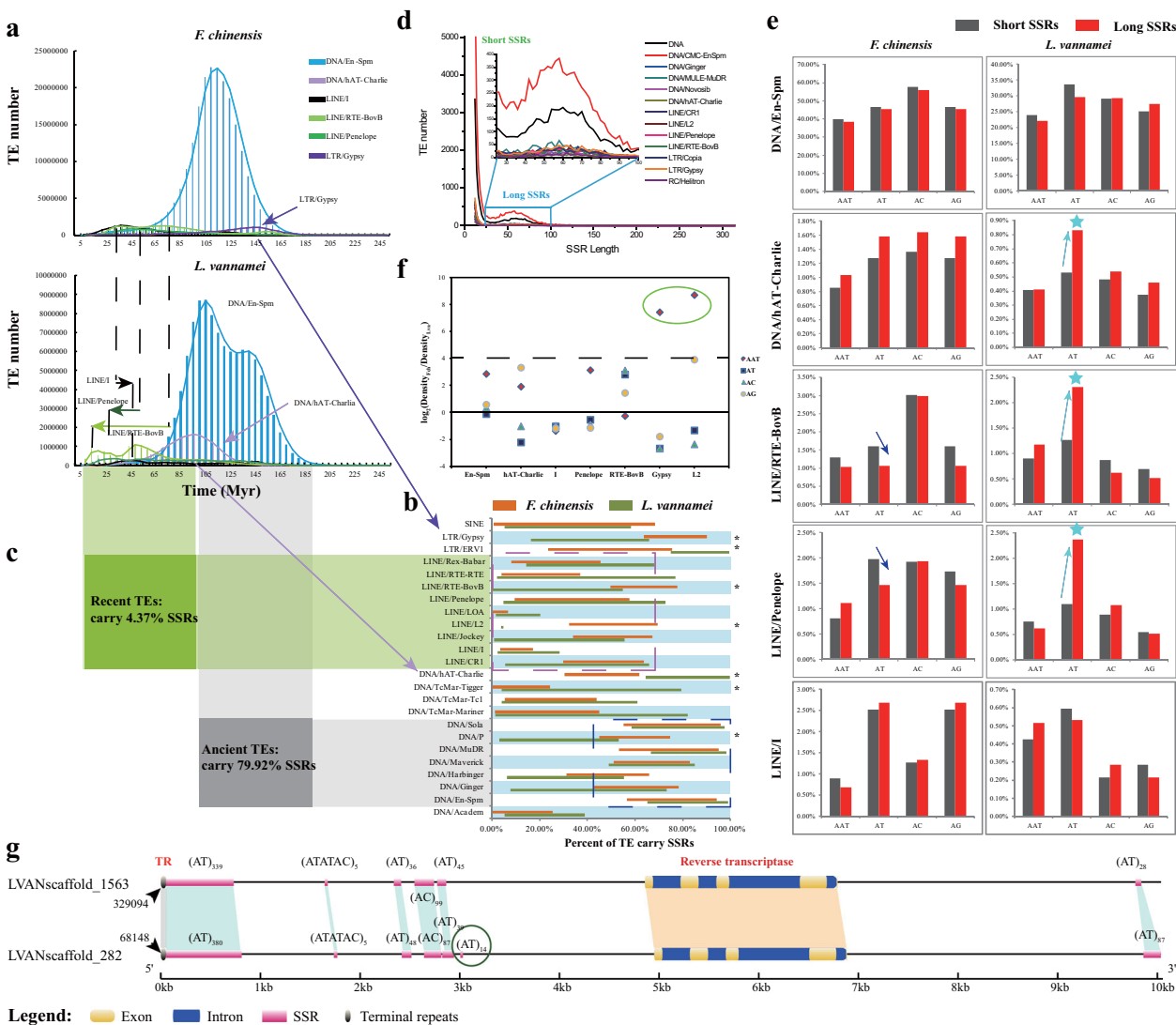

**Fig. 2 Relationship between TEs and SSR expansion. a** Age distribution of major expanded TEs in the two penaeid shrimp genomes. According to previous studies[57], the substitution rate of decapods is $2 \times 10^{-9}$ substitutions per site per year. **b** Percentage of TEs harboring SSRs in the two shrimp genomes. The SSRs located within TEs (hide bar) and 100 bp up- and downstream of TEs (orange and green bars) were counted separately. *Indicates significant difference ($p < 0.05$) of the TEs harboring SSRs between *L. vannamei* and *F. chinensis*. **c** Comparison of ancient and recent TEs that carrying SSRs. **d** Distribution of the number of TEs containing variable lengths of SSRs. Two peaks representing short and long SSRs were identified in the curve plot. **e** Comparison of TEs containg short and long SSRs. The stars indicate the significant differences with $p < 0.05$. The short and long SSRs were selected according to a curve plot of the numbers of TEs with different SSR lengths (Supplementary Fig. 19). Since a single peak (representing short SSRs) was observed in the length distribution of $(AAT)_n$, long $(AAT)_n$ SSRs were considered those with lengths longer than 35 bp. **f** Comparison of SSR density in various TEs of *F. chinensis* and *L. vannamei*. A positive value of $\log_2(\text{Density}_{Fch}/\text{Density}_{Lva})$ indicates that the SSR density in correspondent TE was higher in *F. chinensis* than *L. vannamei*, and verse vera. **g** Synteny of recently transposed TEs (Penelope). SSR elongations and new insertions (green circle) were identified in the transposed TEs.

than 70 bp (Fig. 1f; Supplementary Fig. 15; Supplementary Data: Tables SS3 and SS4), and the longest SSRs reached 13,769 bp and 9212 bp in *L. vannamei* and *F. chinensis*, respectively. Besides, the MMR genes were widely distributed throughout the genomes of both shrimp species, similar to those of other crustaceans, which suggests the presence of functional MMR systems in these penaeid shrimp (Supplementary Fig. 16). Thus, additional mechanisms underlying SSR expansion may exist beside slippage mutation.

Interestingly, we noted that many SSRs were located within or near TEs in these shrimp genomes, which suggesting a potential relationship between SSRs and TEs[2]. Thus, we analyzed compositions of TEs and their surrounding sequences in detail. Similar to that of SSRs, the composition of TEs was comparable

between the two shrimp genomes, and DNA transposons accounted for the majority of the TEs in these genomes (13.00% and 9.33% of the genomes for *F. chinensis* and *L. vannamei*, respectively) (Supplementary Table 10 and Supplementary Fig. 8). These DNA transposons, especially En-Spm, are of ancient origin (~105 Myr) that expanded before the divergence of two penaeids (Fig. 2a). Interestingly, these ancient TEs comprised as much as 79.92% of SSRs in the genome, and furthermore, more than 90% of them were found to harbor SSRs, a significantly higher number than that of recently transposed TEs (<70%) (Fig. 2b, c). Therefore, the dramatic expansion of SSRs in the penaeid shrimp ancestor was tightly associated with a significant expansion of ancient TEs, especially En-Spm DNA transposons.

In addition to these ancient TEs, recently transposed TEs that expanded after penaeid shrimp divergence were also identified to contain SSRs, especially recently expanded species-specific SSRs. Both recently transposed TEs and species-specific expanded TEs were identified in the two penaeid shrimp genomes, and were mainly LINEs, primarily RTE-BovB, Penelope, and LINE/I (Fig. 2a). Although the proportion of recent TEs containing SSRs was low (4.37%), variations were detected between the two shrimp species, especially in RTE-BovB and LINE/L2 (Fig. 2b). Besides, Gypsy and hAT-Charlie elements were found to have specifically expanded in *F. chinensis* and *L. vannamei*, respectively (Fig. 2a; Supplementary Fig. 9 and Supplementary Table 10). Interestingly, almost all the hAT-Charlie elements (99.18%) in *L. vannamei* harbored SSRs, and the proportion was significantly higher than that in *F. chinensis* (61.75%), while the situation was reversed for Gypsy elements (66.13% for *L. vannamei* and 90.27% for *F. chinensis*). Therefore, these TEs may have contributed to the species-specific expansion of SSRs after the divergence of penaeid shrimp.

Unexpectedly, most of the recently and specifically expanded TEs did not specifically harbor the species-specific expanded SSRs ($(AT)_n$ and $(AAT)_n$) except for the Gypsy TEs that specifically contained $(AT)_n$ in *L. vannamei*, and mainly contained $(AAT)_n$ in *F. chinensis* (Supplementary Fig. 17), whereas we found that the differences in SSRs between the two shrimp genomes were the length of $(AT)_n$ and density of $(AAT)_n$ SSRs. *L. vannamei* harbored a greater number of long $(AT)_n$ stretches (>65 bp), while *F. chinensis* had a higher density of $(AAT)_n$ SSRs (Fig. 1f; Supplementary Fig. 15). Unlike that of the other species and other types of SSRs (single peak in length distribution), two peaks, indicating short and long SSRs, were identified in the length distributions of $(AT)_n$, $(AC)_n$, and $(AG)_n$ in the *L. vannamei* and *F. chinensis* genomes (Supplementary Figs. 15 and 18). Importantly, two peaks were also detected in the distributions of the numbers of SSR-containing TEs (Fig. 2d and Supplementary Fig. 19), and the peak representing long $(AT)_n$ SSRs was more obvious in *L. vannamei* than in *F. chinensis*. Interestingly, the recently expanded TEs (Penelope and RTE-BovB) and *L. vannamei*-specific expanded TEs (hAT-Charlie) contained a significantly greater number of long $(AT)_n$ than short $(AT)_n$ in *L. vannamei* ($p < 0.05$), while no difference could be identified among the other types of SSRs or in *F. chinensis* (Fig. 2e), whereas Penelope and RTE-BovB of *F. chinensis* even contained fewer long $(AT)_n$ than short $(AT)_n$ SSRs. Therefore, the recent expansion of TEs was tightly associated with the specific expansion of long $(AT)_n$ SSRs in *L. vannamei*.

Next, we evaluated $(AAT)_n$ density. We found that the recently expanded TEs (LINE/L2) and specifically expanded TEs (Gypsy) contained significantly more SSRs in *F. chinensis* than in *L. vannamei* (Fig. 2a, b). These were the only two types of TEs that specifically harbored $(AAT)_n$ SSRs in *F. chinensis* rather than in *L. vannamei* (Fig. 2f). Gypsy TEs were especially notable, as 45.54% of them harbored $(AAT)_n$ in *F. chinensis*, but only 1.06% in *L. vannamei* (Supplementary Fig. 17). Thus, the expansion of Gypsy seems to have been tightly associated with the specific expansion of $(AAT)_n$ in *F. chinensis*.

We further identified some recently transposed TEs throughout the genome of *L. vannamei*. These TEs (Penelope) showed high similarity in nucleotide sequences (identity >98%) and held complete terminal repeat structures and a gene encoding reverse transcriptase. Both TEs showed high synteny, as did the SSRs located within the TEs, suggesting these SSRs translocated along with the TE (Fig. 2g). Besides, the elongation and newly insertion of SSRs could be detected after TE transposition. Thus, even for the most recently transposed TEs, they are also responsible for SSR expansion.

In summary, the SSR expansion in these two shrimp genomes was tightly associated with TE expansion. A significant expansion of ancient TEs have resulted in an extreme expansion of SSRs in the penaeid shrimp ancestral genome, and the recent expansion TEs was associated with the specific expansion of $(AT)_n$ and $(AAT)_n$ SSRs in *L. vannamei* and *F. chinensis*, respectively.

**High intrachromosomal rearrangement is associated with SSR-rich chromosomal regions in penaeid shrimp.** SSRs are known to contribute to chromosome rearrangement[28,29]; therefore, we investigated level of genome rearrangement in the SSR-rich species *L. vannamei* and *F. chinensis*. To this end, we compared the two shrimp genomes to determine the level of genome synteny. Unexpectedly, these genomes showed a poor synteny even though they had close phylogenetic relationship (Supplementary Fig. 20). Only 293 synteny blocks (involving at least five collinear gene pairs) covering 2149 genes were identified. However, a one-to-one chromosome relationship was clearly displayed, and was further supported by the synteny results for pairwise orthologous genes (Fig. 3a and Supplementary Fig. 21). A total of 12,358 orthologous genes were mostly located within the corresponding chromosomes of the two genomes. However, the intrachromosomal orthologous genes were not collinearly distributed due to chromosomal rearrangements (Fig. 3b).

To further investigate the degree of chromosome rearrangement, we next focused on the *Hox* gene cluster, which is highly conserved across animal genomes and contains at least nine collinearly distributed genes[30]. As expected, the ten *Hox* genes of penaeid shrimp were located on the same chromosome and in the same order as those of other arthropod genomes. However, unlike the conventional compact *Hox* clusters found in many genomes[31], the *Hox* genes of these two penaeid shrimp were widely spaced in the genome (spanning more than 21 Mb in *L. vannamei* and 14 Mb in *F. chinensis*) and interspersed with non-*Hox* genes (Fig. 3c). When comparing the two shrimp genomes, we found that rare non-*Hox* genes were collinearly distributed along with their neighboring *Hox* genes. Thus, intrachromosomal rearrangements had occurred even in a highly conserved gene cluster.

To evaluate if the significant expansion of SSRs in penaeid shrimp genomes may have contributed to the remarkable number of intrachromosomal rearrangements in these species, we determined the SSR and TE content in and around genome rearrangement sites. We validated the candidate rearrangement sites in the *L. vannamei* genome using genome resequencing data from wild and cultured populations. In the regions with a high number of rearrangement sites (Fig. 3d, lower graph), the SSR content was generally low, but sharp peaks of high SSR content were identified adjacent to these regions (Fig. 3d, upper graph). Thus, genomic translocation breakpoints were frequently mapped to SSR-rich chromosomal regions, consistent with a previous report in species with low SSR content[29]. As the potential SSR carriers, TEs displayed similar distributions with that of SSRs, but the sharp peaks of high TE content were generally identified adjacent to the peaks of SSRs (Fig. 3d, upper graph).

**SSRs drive salinity adaptation in penaeid shrimp.** In order to learn the functions of SSRs in the genome, we next studied whether the SSRs in these two species had contributed to their environmental adaptation. We focused on their salinity adaptation capacities, one of the conspicuous phenotypic variations between the two shrimp species. Specifically, the euryhaline Pacific white shrimp *L. vannamei* lives in both coastal and oceanic areas and is capable of surviving in a large range of salinities[32], while *F. chinensis* is naturally distributed in a relative

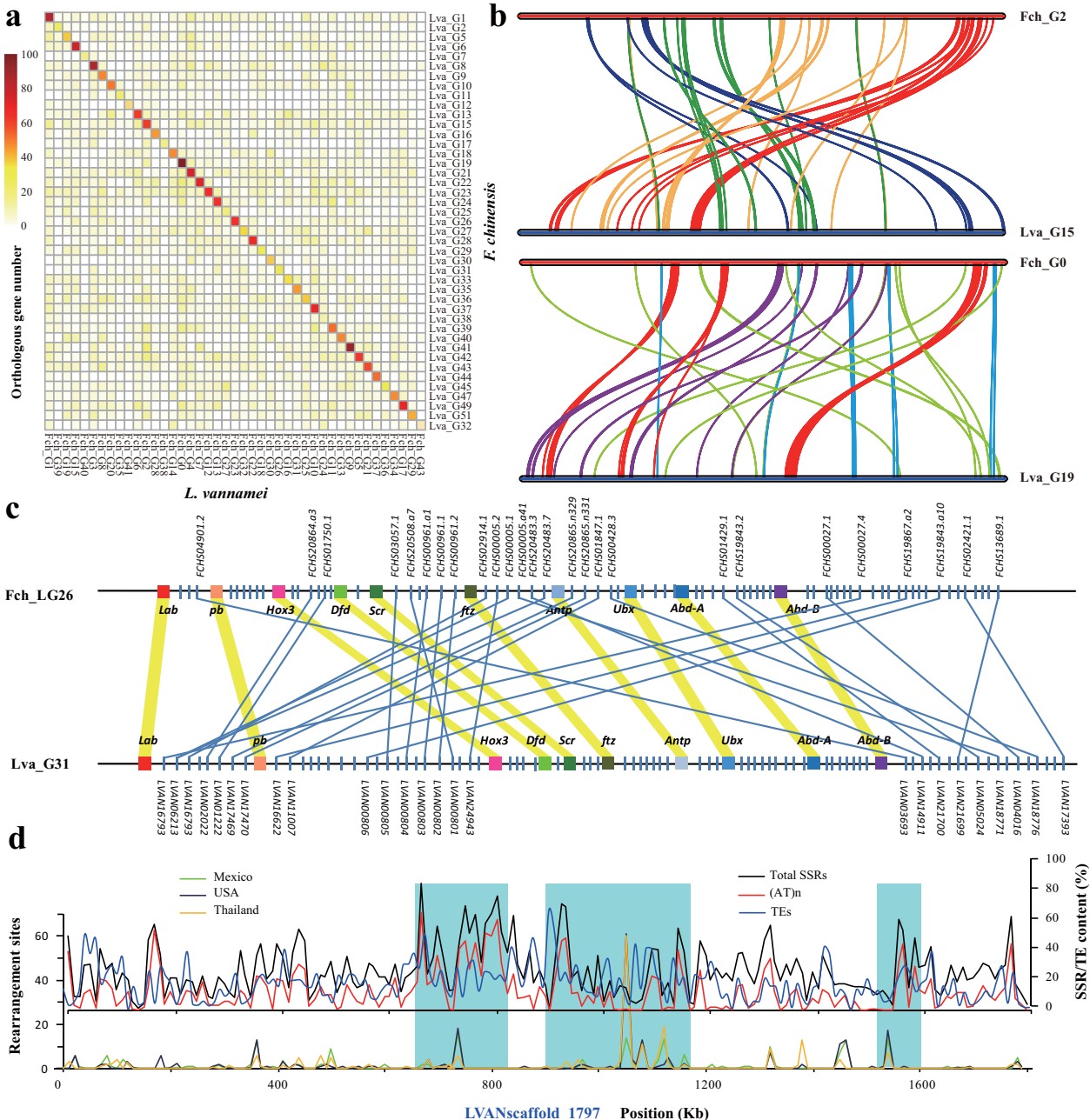

**Fig. 3 Intrachromosomal rearrangement in penaeid shrimp genomes. a** Heatmap of the orthologous gene numbers in each pair of chromosomes from the two shrimp genomes. **b** Intrachromosomal rearrangement between the homologous chromosomes of the two shrimp genomes. **c** Synteny of Hox genes and neighboring genes on the homologous chromosomes of the two shrimp species. **d** Relationship between rearrangement sites and SSRs. The rearrangement sites were calculated according to the paired-end read (Illumina 170 bp libraries) mapping results. When the distance between the paired mapping reads was longer than 50 kb, the site considered a candidate rearrangement site. The number of rearrangement sites in a window of 10 kb was calculated and compared between the shrimp used for genome sequencing and other populations of *L. vannamei* (Mexico, USA, and Thailand). The blue boxes show areas that have high chromosomal rearrangements and low SSR content, but sharp peaks of high SSR content adjacently.

narrow region and cannot be cultivated in freshwater. To identify the genes involved in this differential phenotype, we first performed comparative transcriptomic analyses on these two shrimp species under salinities of 30‰ (control), 9‰ and 3‰ (Supplementary Table 14), and determined the differentially expressed genes (DEGs) (Supplementary Table 15 and Supplementary Fig. 22). For both shrimp species, the DEGs were enriched in pathways related to amino acid and lipid metabolism, including glycine, serine, and threonine metabolism (ko00260), cysteine and methionine metabolism (ko00270), taurine and hypotaurine

metabolism (ko00430), glycerolipid metabolism (ko00561), glycerophospholipid metabolism (ko00564), etc. (Fig. 4a).

Free amino acids and lipids have been shown to be involved in osmoregulation in decapods, including many other penaeid shrimp species[33–35]. We performed metabolome comparisons under low-salinity stress in both shrimp species. As expected, many metabolites related to amino acid and lipid metabolism, which may be involved in osmoregulation in decapods, were differentially regulated, and the functionally enriched pathways were very similar to those identified by the DEG analysis

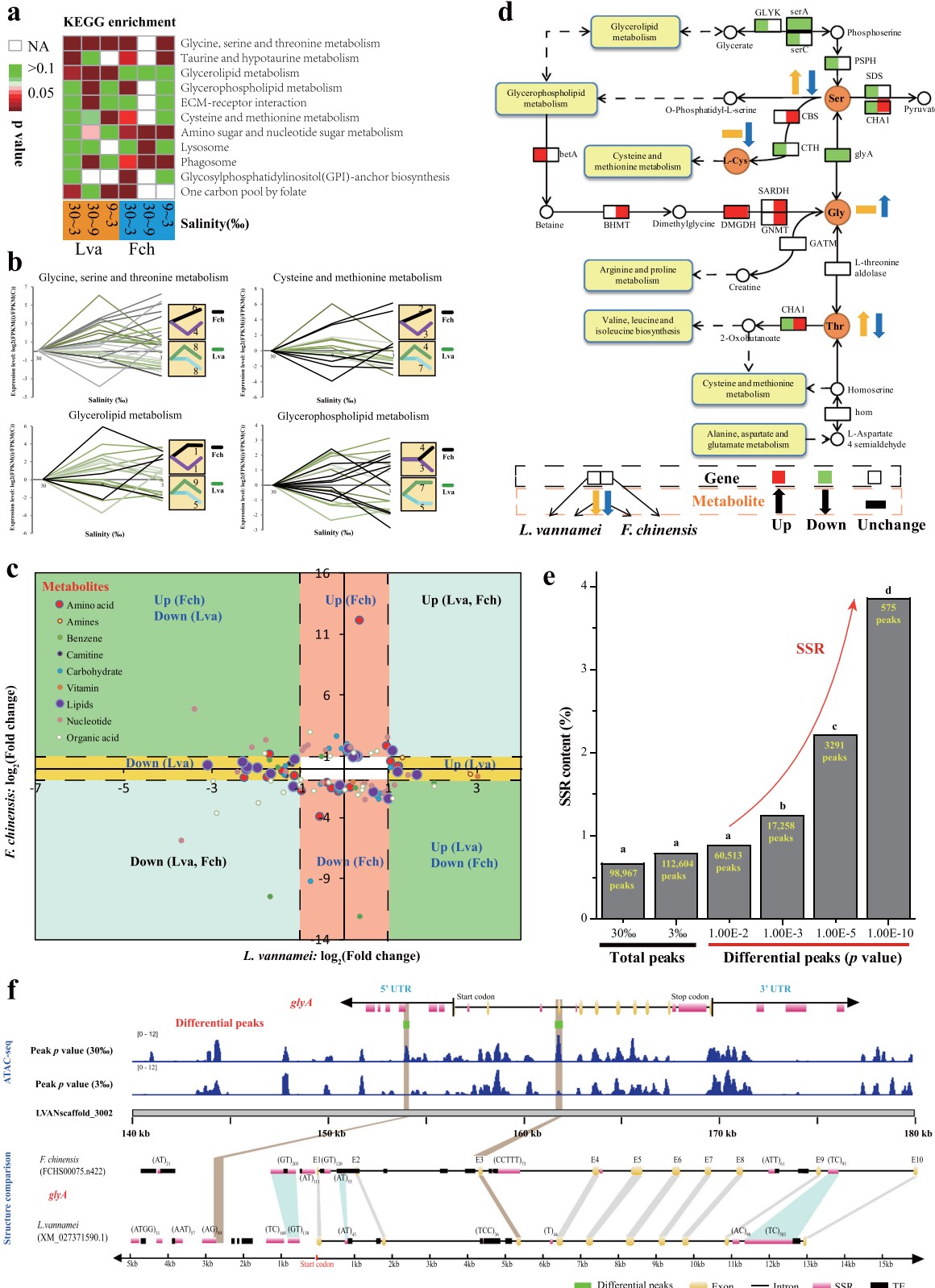

(Supplementary Data: Tables SS5–SS7 and Supplementary Fig. 23).

Beside similarity, the two shrimp species also displayed some differences in salinity adaptation. Although similar number of genes in the pathways related to amino acid and lipid metabolism were identified in the genomes of the two species, more DEGs were identified in *L. vannamei* than in *F. chinensis*

(Supplementary Tables 16 and 17). Furthermore, these DEGs displayed apparently differential expression patterns between the two species (Fig. 4b). In *L. vannamei*, the DEGs were mostly differentially expressed at the initial change in salinity (from 30‰ to 9‰), but in *F. chinensis*, they were mostly differentially expressed when the salinity decreased from 9‰ to 3‰. These data indicate that *L. vannamei* responds to low salinity more

**Fig. 4 Multi-omic analyses of two penaeid shrimp species under low-salinity stress. a** KEGG enrichment analysis of the DEGs of *L. vannamei* (Lva) and *F. chinensis* (Fch). Only the pathways with significantly enriched DEGs ($p < 0.05$) are shown on the heatmap. NA indicates no DEGs identified in related pathways. **b** Expression patterns of the DEGs in different pathways. The number of DEGs and their corresponding expression patterns are shown in the yellow box at right. **c** Differentially regulated metabolites in penaeid shrimp under low-salinity stress. Metabolite levels were compared between the two shrimp species, and the metabolites were divided into four groups, namely, those differentially regulated in only *L. vannamei* (yellow background), those differentially regulated in only *F. chinensis* (orange background), those up- or downregulated consistently in both shrimp species (light blue background), and those with different regulation patterns between the two shrimp species (green background). A white background indicates no significant difference. **d** Correlations between DEGs and differentially regulated metabolites in the pathway of glycine, serine, and threonine metabolism in both the *L. vannamei* and *F. chinensis* genomes. **e** SSR contents in all identified peaks at 3% and 30% salinity and only the differential peaks (3% vs. 30%) identified by ATAC-seq. The differential peaks were identified according to various $p$ values from differential analyses of ATAC-seq. **f** ATAC peak distribution and structure comparison of the *glyA* gene in both *L. vannamei* and *F. chinensis*. The brown lines indicate orthologous regions of the differential peaks in the genes.

rapidly than *F. chinensis*, and thus these responses might endow *L. vannamei* with a better osmoregulatory capacity.

Similar to the RNA-seq results, few metabolites displayed similar regulation patterns between the two shrimp species. Most metabolites were up- or downregulated in one shrimp species, but unchanged in the other one (Fig. 4c). As indicated by the KEGG map, the glycine, serine, and threonine metabolic pathway was potentially co-regulated with many other amino acid and lipid metabolic pathways (Fig. 4d and Supplementary Fig. 22). Supportingly, the DEGs in these pathways displayed similar expression patterns (Fig. 4b). The metabolite levels were tightly correlated with gene expression (Supplementary Fig. 24). For example, the downregulation of several genes (e.g. *PSPH*, *SDS*, *CHA1*, and *CTH*) corresponded to a high serine level upregulated *L. vannamei*, but the upregulation of *CHA1* and *CBS* corresponded to a low serine level in *F. chinensis* (Fig. 4d). Similar phenomena were also detected for the production of glycine, L-cysteine, and threonine. These results suggest that the DEGs identified in the transcriptome analysis and their expression patterns directly affect metabolite regulation, which would result in variations in the osmoregulatory capacities of these two penaeid shrimp species and hence their different salinity adaptation capacities.

Next, we compared the structures of the orthologous genes in the pathway of glycine, serine, and threonine metabolism of these two shrimp species to investigate if the different SSR distribution in these species could underlie the differential gene expression observed under different salinity conditions in these two species. The composition of the SSRs located within introns or UTRs showed significant differences in most orthologous genes, even though exons displayed high synteny between the two shrimp species (Supplementary Fig. 25). A high number of new SSRs have been inserted or deleted in these genes, including some extraordinarily long SSRs, e.g., $(AT)_{676}$ in the intron 2 of *GNMT* and $(AT)_{637}$ in the intron 2 of *PSPH*. In addition, some SSRs were significantly elongated in one shrimp species compared to the other despite being located at the same position in an orthologous gene, e.g., $(AG)_n$ in the intron 5 of *DMGDH* had the motif repeat numbers of 53 and 421 in *F. chinensis* and *L. vannamei*, respectively. Even for SSR-free genes (e.g., *Klf1*), the SSRs located in the UTRs also showed some differences (Supplementary Fig. 25). Therefore, the compositions of the SSRs between pairs of orthologous genes in these two shrimp species displayed significant differences, which could potentially affect the regulation of gene expression.

To assess if SSRs could indeed affect gene expression, we performed ATAC-seq to identify the regulatory regions that responded to low-salinity stress (Supplementary Tables 17 and 18 and Supplementary Fig. 26). As many as 3291 and 4294 differential ATAC peaks ($p < 1.00E-5$), representing potential regulatory regions responding low-salinity stress, were identified by comparing the control (30‰) and low-salinity (3‰) samples

of *L. vannamei* and *F. chinensis*, respectively (Supplementary Table 19). In total, 762 and 2226 genes were located within or around these differential ATAC peaks in *L. vannamei* and *F. chinensis*, respectively. These genes were mostly involved in various pathways related to biosynthesis and metabolism, including pathways related to amino acid and lipid metabolism (Supplementary Figs. 27 and 28), which was consistent with the RNA-seq results. Thus, the chromatin accessibility of these genes was adaptively altered in response to low-salinity stress. Analysis of the composition of accessible regions showed that the SSR content was significantly higher in the differential ATAC peaks (peaks with differential test $p < 1.00E-3$) than in the normal peaks (total 98,967 and 112,604 peaks under the salinity of 30‰ and 3‰) (Fig. 4e). Furthermore, the SSR content in the differential peaks increased following the significance level of the peaks, and even reached 3.86% in the differential peaks with the significance $p$ value $<1.00E-10$, which was significantly higher than that in the normal peaks (0.66%, 30‰ salinity). Similar to that of SSRs, the content of TEs in the differential peaks increased as the significance level increased, but unchanged until $p$ value $<1.00E-5$ (Supplementary Fig. 29). The accessible regions with greater significance ($p$ value) of differential peaks indicating that these regions were more active and more important in response to environmental stresses. Therefore, SSRs and TEs as located in the regulatory regions might play important roles in regulating gene expression under low-salinity stress.

To specifically understand how SSRs might affect expression of genes involved in amino acid metabolism, we focused on *glyA*, which encodes serine hydroxymethyltransferase that regulates the production of both glycine and serine, and it is a key enzyme in the glycine, serine, and threonine metabolic pathway (Fig. 4d). Two differential peaks were identified in the 5′ UTR and exon 3 of *glyA*, and the differential peak in the 5′ UTR partially overlapped with an SSR $((AG)_{89})$ in *L. vannamei* (Fig. 4f), whereas the corresponding SSR was not present in the *glyA* orthologue of *F. chinensis*. In addition, an SSR insertion $((CCTTT)_{73})$ was also identified near the exon 3 in *F. chinensis*, which might potentially affect its transcription. In comparison, TEs did not show obvious correlation with these two differential peaks.

By combining the above results, it can be concluded that SSRs play important roles in regulating gene expression, and help regulate osmoregulation in penaeid shrimp.

## Discussion

As SSRs make up a large portion of the penaeid shrimp genome, we aimed to determine whether SSRs drive genome plasticity and adaptive evolution in these species. Many factors have been found to drive genome plasticity, and TEs are one of the most widely investigated factors, since they make up a large part of the genome[36]. However, in this study, we found that SSRs are the major type of DNA repeats in penaeid shrimp genomes and shape the genome plasticity to allow for an increased ability to adaptively

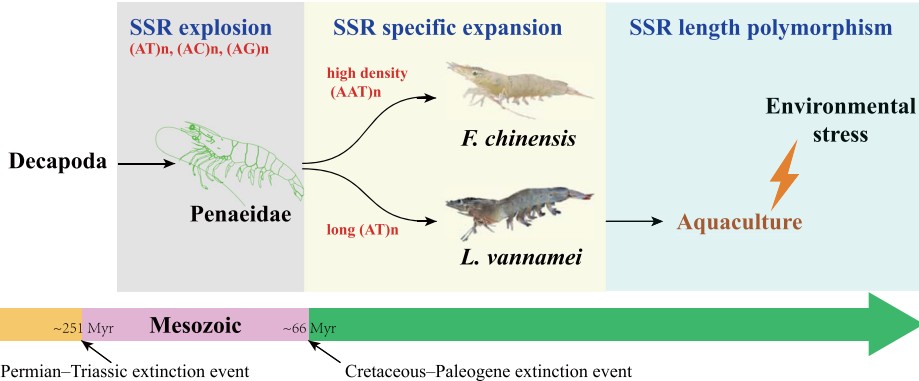

**Fig. 5 SSR expansion during penaeid shrimp evolutionary history.** Timescale of SSR expansion and relationships with the origination and divergence of penaeid shrimp.

evolve. A massive expansion of SSRs occurred in the penaeid shrimp ancestor, and an additional specific SSR expansion occurred after the divergence of this group (Fig. 5). Newly generated SSRs were also identified in some recently transposed TEs and orthologous genes (Fig. 2g and Supplementary Fig. 25). In addition, the SSR length was highly polymorphic among different aquaculture populations of penaeid shrimp (Supplementary Fig. 30). Therefore, we suggested that SSRs have contributed to the high genomic diversity of penaeid shrimp genomes from ancient times to the present. Further, many intrachromosomal rearrangement events and significant gene structure alterations were found to be caused by SSR expansion. Chromosomal rearrangements leading to genome plasticity have been identified in some species, and the high incidence of repetitive sequences has been suggested as a potential driving force for this phenomenon[22,37]. In addition, tandem DNA repeats, including SSRs, have been identified as hot spots for recombination since translocation breakpoints are preferentially allocated in repeat-rich regions[29,38]. The simple base composition of SSRs provides an important opportunity for pairing to produce segmental duplications and genome rearrangements during replication. Significant alterations in gene structure (SSR elongation and insertion) were observed in most of the orthologous genes of the two shrimp genomes (Supplementary Fig. 25). SSR expansions and/or contractions in protein-coding regions can lead to a gain or loss of gene function, and SSRs located in UTRs and introns can affect gene transcription, mRNA splicing, or export to cytoplasm[12]. Thus, SSRs are a significant source of genomic plasticity and can alter genomic structures and gene transcription.

As an important component of the genome, TEs are also important for genomic plasticity[22]. TEs may also contribute to the genome rearrangements, and gene structure and function regulation. Whereas, in the penaeid shrimp genomes, TEs were identified to be associated with SSR expansion, similar with that of micron and RTE retrotransposons[39,40]. Both ancient (DNA transposons) and recently transposed TEs (DNA retrotransposons) functions as carriers for SSR expansion in the penaeid shrimp genomes. TEs were generally identified adjacent to SSRs in the genome (Fig. 4f). In contrast to slippage mutation-based self expansion, the TE harboring may be a new and more efficient way for the SSR expansion in SSR-rich species.

SSRs were identified in some recently transposed retrotransposons (Fig. 2g), indicating that the expansion of SSRs along with retrotransposons is active at present, and will remain active in the future. Beside TE carrying, the new generation of SSRs that has escaped from the MMR system was also identified in the penaeid shrimp genomes, as indicated by the recently transposed TEs and SSR polymorphisms in different populations (Fig. 2g and

Supplementary Fig. 30). Mutations that escaped from the MMR system will become new alleles at the SSR loci, and then regulate and/or change gene products, eventually leading to phenotypic changes[12]. Thus, vast numbers of SSRs have been retained from the penaeid shrimp ancestor, and newly generated SSRs will continue to contribute to the adaptive evolution of these shrimp.

Genomic plasticity shaped by repeats enables organisms to adapt to environmental changes and occupy novel niches[41]. Adaptive radiation and rapid diversification were generally occur after mass extinction events, as more space and opportunities are available at such times[42]. In terms of evolution, decapods are important members of the modern evolutionary fauna after mass extinction[43]. The divergence of Decapoda was consistent with the time after the Permian–Triassic extinction (Fig. 5). Harsh conditions led to the extinction of approximately 96% of marine species[44], followed by a radiation of many new life forms, including shrimp-like decapods, in the Early-Middle Triassic[45]. Another radiation event occurred in the Late Cretaceous, generating many modern penaeid shrimp species[25,46]. During these two events, an extreme expansion and species-specific expansion of SSRs were identified in penaeid shrimp genomes, suggesting the potential relationship between SSR expansion and shrimp adaptive evolution. Supportingly, some Penaeidae-specific and species-specific expanded gene families were identified during these two stages. The remarkable genomic plasticity shaped by SSR expansions may have helped penaeid shrimp adaptively evolve for rapid diversification after mass extinctions.

In addition to their evolutionary roles, SSRs play important functional roles in environmental adaptation. The influence of SSRs on gene regulation, transcription and protein function typically depends on motif repeat number[3]. SSR expansions or contractions in protein-coding regions can lead to a gain or loss of gene function via frameshift mutations, while SSR variations in UTRs could regulate gene expression by affecting transcription and translation[12]. Significant variations in SSR composition, including SSR insertion, deletion, expansion, and contraction, were identified in the orthologous genes of the two shrimp species, and could ultimately affect the functions of these genes. SSRs located in regulatory regions can regulate gene expression, and cause variations in the osmoregulatory capacities of the two shrimp species.

In this study, we identified the potential mechanisms of SSR expansion, and their evolutionary and functional roles during adaptive evolution, which provide a new view of the ways for SSR expansion and make SSRs as new key elements for the genome evolution in SSR-rich species. However, further investigations of SSRs are needed to study the SSR origination and functions in SSR-rich species, as many problems are still unsolved. How did

TEs carry SSRs? Why are dinucleotide SSRs specifically selected by penaeid shrimp? What other functions do SSRs serve in SSR-rich species? Further studies of SSRs will help us understand the penaeid shrimp genomes better and extend its utilization in many more scientific areas. For example, SSR length polymorphisms have been shown to play important roles in viral resistance in various aquaculture populations of *L. vannamei*[47]. As the most important economic shrimp group all over the world, penaeids are widely cultured, and future utilization of the potential functions of SSRs will promote the selective breeding of these organisms.

## Methods

**Genome sequencing**. The muscle of a wild male adult of *F. chinensis* was collected from the sea area of Jimo Qingdao for DNA extraction and genome sequencing. Total genomic DNA was extracted using TIANamp Marine Animal DNA Kits (Tiangen, Beijing, China). Paired-end and mate-paired libraries with insert sizes ranging from 137 to 17 kb were constructed according to the instructions provided in the Illumina library preparation kit (Illumina, San Diego, USA). The constructed libraries were then sequenced on the Illumina HiSeq 2500 platform (Illumina, San Diego, USA). After removal of the Illumina sequence adaptors and low-quality reads with Trimmomatic v.0.35 (http://www.usadellab.org/cms/index.php?page=trimmomatic), the retained clean reads were used for subsequent analyses.

For PacBio library construction, at least 10 μg of genomic DNA was sheared to ~20 kb. Fragments below 7 kb were filtered out using BluePippin (Sage Science, Beverly, USA). SMRTbell libraries were constructed for single-molecule real time (SMRT) sequencing using the P6C5 sequencing chemistry (Pacific Biosciences, San Diego, USA), and then sequenced on the PacBio RSII sequencing platform (Pacific Biosciences, San Diego, USA).

For Hi-C library construction, muscle samples from both *F. chinensis* and *L. vannamei* were flash frozen and pulverized prior to formaldehyde crosslinking. Then, restriction enzyme (*Mbo*I/*Dpn*I/*Hind*III) digestion was used for cell lysis, and the ends were labeled with biotin and then supplemented and connected. Protease K and SDS were added to remove the cross-links, and the DNA was purified and extracted by AMPure XP beads. The biotin-labeled DNA fragments were captured with M-280 Streptavidin beads. The ends were repaired by adding A-tails, and then, the sequencing connector was attached. Finally, the constructed library was used for sequencing on the Illumina X-TEN platform (Illumina, San Diego, USA) with 2 × 150 bp reads.

**Genome size estimation**. The genome size of *F. chinensis* was estimated by flow cytometry and K-mer frequency. The hemolymph of adult *F. chinensis* was collected by chopping with a razor blade in PBS buffer, and the homogenized cell suspension was filtered through a 30 μm nylon filter. Then, 12 μL of propidium iodide (50 mg/mL) and the cells were stained with 2 μL of RNase (10 mg/mL) for 20 min for measurement in a BD FACSAria II (BD Biosciences, San Jose, USA). The genome size estimated by flow cytometry was compared with the reference standard (mouse: *Mus musculus* "KM", genome size ~2.50 Gb). Jellyfish was used to calculate K-mer frequencies and then estimate the genome size with the high-quality reads from the short-insert libraries[48]. The genome size was estimated as the ratio of the total number of K-mers from all reads to the peak depth. Since the genome size estimated by these two approaches was similar, we used the K-mer frequency estimate to represent the genome size of *F. chinensis*.

**Genome assembly**. The *F. chinensis* genome was de novo assembled using WTDBG software based on the subreads from PacBio sequencing[49]. The assembled sequence was then polished using Quiver (SMRT Analysis v2.3.0) with the default parameters. Besides, paired-end Illumina clean reads from 137, 170, 300, and 500 bp libraries were also used for iterative error correction several times. Finally, scaffolds were generated through gap-filling with SSPACE 3.0 with the parameter values "-x 1 -m 50 -o 10 -z 200 -p 1" using meta-paired Illumina sequencing reads (2, 5, 10, 12, and 17 kb libraries).

To assemble the chromosome-level genomes of *F. chinensis* and *L. vannamei*, the Hi-C sequencing data were aligned to the assembled scaffolds by using Juicer[50]. The Hi-C data were independently analyzed in the HiC-Pro pipeline (default parameters and LIGATION_SITE = GATC). The scaffolds were clustered onto chromosomes with the 3D-DNA pipeline (version 180419)[51], and the contact maps were plotted using HiCPlotter software[52].

**Quality assessment of genome assembly**. To evaluate the quality of the genome assembly, the Illumina sequencing reads from the paired-end library (170 bp) were mapped using Bowtie2 (ref. [53]). A total of 91.12% of the reads could be mapped to the current assembly, which covered 82.46% of the genome (Supplementary Table 5). The unmapped genomic regions were mostly composed of SSRs[2]. To evaluate the completeness of the genome assembly in the gene regions, we mapped the unigenes from the transcriptome data to the shrimp genome. A total of 64,708 unigenes were assembled using Trinity[54], and then mapped to the scaffold using BLAT[55]; 94.83% of the unigenes could be identified in the assembly (Supplementary Table 6). In addition, we used BUSCO to evaluate the quality of the genome assembly (http://gitlab.com/ezlab/busco). The 1066 conserved BUSCOs of Arthropoda were used as the database for a BLAST search. A total of 92.68% of conserved genes could be identified in the *F. chinensis* genome (Supplementary Table 7).

**Repetitive sequence annotation**. For TE annotation, both RepeatModeler and RepeatMasker were used to perform de novo identification[56]. The substitution rates of TEs were calculated between the genome and repeat consensus sequences. The divergence times of TEs was calculated based on the TE substitution rate according to a previously determined molecular clock of decapods ($2 \times 10^{-9}$ substitutions per site per year)[57]. The SSR annotation was conducted by SciRoKo v3.4 (ref. [58]). The average length and density (number per Mb) of the total SSRs and each type of SSR were calculated and compared with those of other species.

**Protein-coding gene prediction and annotation**. Protein-coding genes were predicted through a combination of homology-based prediction, de novo prediction, and transcriptome-based prediction methods. For homology-based prediction, homologous proteins from *L. vannamei*, *Daphnia pulex*, *P. virginalis*, *Parhyale hawaiensis*, *Eulimnadia texana*, *Drosophila melanogaster*, *Anopheles gambiae,* and *Bombyx mori* were downloaded from NCBI and mapped against the *F. chinensis* genome with Exonerate version 2.2.0 (http://www.ebi.ac.uk/~guy/exonerate/). For de novo prediction, the coding regions were predicted using Augustus v2.5.5 (ref. [41]). For transcriptome-based prediction, the RNA-Seq data were mapped against the assembly using Tophat v2.1.1, and then, the transcripts were converted to gene models using Cufflinks v2.2.1 (ref. [59]). Finally, all the gene models derived from the above three methods were integrated with EVidenceModeler (EVM)[60].

BLASTP against the NR and SwissProt databases was used for functional annotations of the predicted genes with an *E*-value of 1E$^{-05}$. InterProScan and HMMER were used for protein domain annotation by mapping to the InterPro and Pfam databases[61,62]. KEGG Automatic Annotation Server (KAAS) was used to annotate the pathways in which the genes might be involved through mapping against the KEGG database. The Gene Ontology (GO) classifications of the genes were extracted from the corresponding InterProScan or Pfam results.

Functional enrichment analysis was conducted on a subset of interesting genes according to their GO and KEGG classifications. The enriched GO terms and KO pathways were calculated relative to the background of all protein-coding genes using Omicshare CloudTools (http://www.omicshare.com/).

**Gene family analyses**. Gene family clustering was performed using OrthoMCL[63]. All-to-all blast searches using the BLASTP program were conducted on the protein-coding genes of 17 arthropods, including *F. chinensis*, *L. vannamei*, *P. virginalis*, *E. sinensis*, *P. hawaiensis*, *D. pulex*, *E. texana*, *Eurytemora affinis*, *Tigriopus californicus*, *Strigamia maritima*, *Ixodes scapularis*, *P. humanus*, *Tetranychus urticae*, *Acyrthosiphon pisum*, *Locusta migratoria*, *B. mori*, and *D. melanogaster*. We used CAFE software for computational analysis of gene family evolution[64], and then identified the expansion and contraction of gene families. Expanded gene families sequences were aligned using the program MAFFT version 5 (ref. [65]). Maximum likelihood (ML) analyses were conducted using RaxML[66]. The divergence time of the expanded gene families was estimated by BEAST v2.6.3 using Relaxed Clock Log Normal model[67]. The divergence time of *P. virginalis* (~225 Myr) was set as the calibration information.

**Phylogenetic analysis**. A total of 51 single-copy homologous genes in the 17 species of Arthropoda were extracted from the above gene family analysis and used for phylogenetic tree construction. The amino acid sequences of the single-copy genes were fully aligned using MUSCLE 3.6 with the default settings[68]. Positions with gaps and missing data were trimmed using an in-house Python script (allfasta2snp.py, https://github.com/jianbone/L_vannamei_genome). The final dataset containing 8232 amino acids was used to construct the phylogenetic tree. Afterward, the conserved alignments of the single-copy genes were concatenated to form the final alignment matrix. Then, the ML method was used for phylogenetic analysis using RAxML under the JTT matrix-based model[66]. The initial trees for the heuristic search were obtained automatically by applying the neighbor-joining and BIONJ algorithms to a matrix of pairwise distances estimated using a JTT model. The divergence time was estimated based on the phylogenetic tree. A time-calibrated phylogeny was inferred using a relaxed molecular clock method as implemented in BEAST v2.6.3 (ref. [67]). Chains were run for 10,000,000 generations, and runs were sampled every 1000th generation. The initial 10% were discarded as burn-in. Divergence time calibration information was obtained from the TimeTree database (http://www.time.org/).

**Genome synteny analysis between two penaeid shrimp species**. The orthologous genes between the two shrimp species, *F. chinensis* and *L. vannamei*, were identified by using reciprocal best hit, in which orthologs are assumed if two genes in different genomes find each other as the best hits in the other genome. Synteny plot diagrams were drawn and compared using the Python jcvi library. Synteny

blocks with at least five collinear orthologous genes were detected using MCSCANX software[69] with the following standard parameters: MATCH_SCORE: 50, MATCH_SIZE: 20, GAP_PENALTY: -1, OVERLAP_WINDOW: 5, E_VALUE: 1e-05 and MAX GAPS: 25. Dotplots were drawn and compared using Multi-Genome Synteny Viewer[70].

A total number of 12,618 orthologous genes were identified between the two shrimp species, and positively selected genes were identified using yn00 in the PAML package. Genes with the $d_N/d_S$ value larger than 1 were identified as under positive selection.

**Transcriptome sequencing and analyses**. To investigate the mechanisms of osmoregulation in response to low-salinity stress, transcriptome sequencing and comparative analyses were performed on the two penaeid shrimp species with the help from GENE DENOVO. Adult shrimp of *F. chinensis* (6.8 ± 0.4 cm) and *L. vannamei* (7.2 ± 0.5 cm) were collected at the shrimp culture laboratory of the Institute of Oceanology Chinese Academy of Sciences (IOCAS) in Qingdao, and acclimated to a salinity of 30‰. The salinity was gradually reduced to 3‰, and the animals were allowed to acclimate to salinities of 9‰ and 3‰ for 24 h. Then, hepatopancrea samples were collected from animals acclimated to salinities of 30‰, 9‰, and 3‰. According to the standard manufacturer's protocol, total RNA was isolated and purified from the samples using TRIzol extraction reagent (Thermo Fisher Scientific, USA). RNA quality and concentration were determined by 1% agarose gel electrophoresis, and RNA concentration was assessed using a Nanodrop 2000 spectrophotometer (Thermo Fisher Scientific, USA). Transcriptome libraries were prepared according to the instructions of the TruSeq RNA Library Prep Kit (Illumina, San Diego, USA), and then sequenced on the Illumina HiSeq 2500 platform. The TopHat v1.2.1 package was used to map the transcriptome reads to the shrimp genomes[59]. Then, fragments per kilobase of transcript per million fragments mapped (FPKM) was calculated using Cufflinks v2.2.1 (http://cole-trapnell-lab.github.io/cufflinks/). The differential gene expression analysis was conducted by using edgeR V3.10 (ref. [71]).

**Metabolome sequencing and analyses**. In order to investigate whether free amino acids and lipids were associated with osmoregulation in penaeid shrimp, metabolome sequencing and analyses were performed on the two shrimp species under low-salinity stress. The samples collected from the shrimp under salinities of 30‰ and 3‰ were used for metabolite extraction. Each sample (50 mg) was homogenized with 1000 μl of ice-cold methanol/water (70%, v/v). Cold steel balls were added to the mixture, which was then homogenized for at 30 Hz for 3 min. The mixture was agitated for 1 min and then centrifuged at 12,000 r.p.m. at 4 °C for 10 min. The collected supernatant was used for LC-MS/MS analysis by using LC-ESI-MS/MS system (UPLC, Shim-pack UFLC SHIMADZU CBM A system; MS, QTRAP® System). LIT and triple quadrupole (QQQ) scans were acquired on a triple quadrupole-linear ion trap mass spectrometer (QTRAP), The QTRAP® LC-MS/MS System, equipped with an ESI Turbo Ion-Spray interface, operating in positive and negative ion mode and controlled by Analyst® 1.6.3 software (https://sciex.com/products/software/analyst-software). Instrument tuning and mass calibration were performed with 10 and 100 μmol/l polypropylene glycol solutions in the QQQ and LIT modes, respectively.

Orthogonal partial least-squares discriminant analysis (OPLS-DA) was used to identify the differential metabolites[72]. Two criteria were used to assess the significance of differential metabolite levels. One was the fold change of the variable, and the other was the variable importance in projection (VIP) value. When its fold change was ≥2 or ≤0.5, and VIP was ≥1, the metabolite was considered differentially regulated. The VIP value was calculated using OPLSR. Anal of MetaboAnalystR from R package[73]. The data were log transformed (log2) and mean centered before OPLS-DA was performed.

**ATAC-seq sequencing and analyses**. To identify the regulatory regions related to low-salinity adaptation in penaeid shrimp, an assay for transposase accessible chromatin with high-throughput sequencing (ATAC-seq) was performed on the samples taken under salinities of 30‰ and 3‰. The samples were spheroplasted prior to incubation with Nextera Tn5 Transposase (Illumina, San Diego, USA). After the transposition reaction, DNA purification, and PCR amplification, the libraries for sequencing were prepared according to the protocols of ATAC-seq[74]. ATAC-seq and the Nextera workflow were designed for sequencing using Illumina high-throughput sequencing instruments (Illumina, San Diego, USA). The sequencing reads were mapped to the shrimp genomes using Bowtie2 (ref. [53]). Peak calling was performed using MACS v2.1.0 with a *p* value cutoff of $1E^{-05}$ and the broad flag and were filtered based on mappability[75]. Peaks appearing in common in the replicate samples were called and used for the following comparative analyses. UCSC Genome Browser was used for the visualization of the peaks. DESeq2 was used for differential peak identification with nominal *p* values. The relationships of the differential peaks to the annotated genes were calculated according to the positions of the genes.

**Statistics and reproducibility**. The statistics for this study are conducted using Student's *t*-test (between two groups) and one-way ANOVA (among three or more groups) using SPSS 22.0 software (https://www.ibm.com/analytics/spss-statistics-software). Significant differences are indicated when *p* value < 0.05. We did

transcriptome, metabolome, and ATAC sequencing with three replicates for each condition.

**Reporting summary**. Further information on research design is available in the Nature Research Reporting Summary linked to this article.

## Data availability

Genome sequence data that support the findings of this study have been deposited in NCBI GenBank with the accession number of JABKCB000000000 and BioProject accession codes of PRJNA627295. The *F. chinensis* genome and the annotation information can also be downloaded from the Shrimp Genome Database (http://www.genedatabase.cn/fch_genome.html).

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

## Acknowledgements

We acknowledge financial support from National Key Research & Development Program of China (2018YFD0900103 to Xiaojun Zhang), the National Natural Science Foundation of China (41876167 to J.Y. and 31830100 to F. Li), the Senior User Project of RV KEXUE (KEXUE2018G19 to F. Liu and Xiaojun Zhang), the grants from Qingdao National Laboratory for Marine Science and Technology (MS2017NO04 to F. Li), the China Agriculture Research system-48 (CARS-48 to F. Li). We acknowledge the support from High Performance Computing Center, Institute of Oceanology, CAS. We would like to express our gratitude to Prof. Chengzhi Liang and Dr. Qiang Gao (Institute of Genetics and Developmental Biology, Chinese Academy of Sciences) for their support of the shrimp genome assembly. We appreciate Dr. Hao Huang (Hainan Guangtai Ocean Breeding Co., Ltd) for the help with shrimp materials.

## Author contributions

J.X., F. Li, L.F., J.Y. and Xiaojun Zhang initiated, managed, and drove the shrimp genome sequencing project. Xiaojun Zhang, C.Z., and J.K. collected the animal material. J.Y., Xiaojun Zhang., M.W., Y.S., and G.F. prepared DNA sequencing and analysis. J.Y., Y.S., and M.W., performed genome assembly, gene annotation, genome structure analyses, and phylogenetic analyses. J.Y., Xiaojun Zhang. and Y.S. conducted SSR analysis and genome synteny analysis. J.Y., Xiaojun Zhang, F. Liu, and Xiaoxi Zhang conducted RNA-seq, metabolome, ATAC-seq sequencing, and analysis. Y.Y. and Y.S. conducted the population genetics analysis. C.L., J.Y., and Y.S. constructed the shrimp genome database and submitted the genome data. F. Li, J.X., J.Y., Xiaojun Zhang, and Y.S. wrote the manuscript and additional supplementary files. L.F., M.W., S.L., Y.G., and J.K. revised the manuscript. All authors read and approved the final manuscript.

## Competing interests

The authors declare no competing interests.
