## [Peer Review File · Communications Biology]

Reviewers' comments:

Reviewer #1 (Remarks to the Author):

This manuscript explores the role of simple sequence repeats (SSRs) in the genomes of two penaeid shrimp species. The authors have done a very comprehensive analysis based on genome resequencing, de-novo sequencing, transcriptomics and metabolomics to shed light on how these repeats have evolved and how they contribute to the adaptation of shrimp species to their specific ecological niche. The main outcome of this study is that the repeats appear to drive adaptive evolution by increasing genome plasticity with consequences for the evolution of phenotypes that convey a fitness advantage under given environmental conditions. Overall, this appears to be a thoroughly conducted study encompassing very different approaches and methods to reveal the significance of repeats for the evolution of shrimp species. I was impressed by the amount of work the authors have put into this study, and their manuscript was carefully written up. Generally, results have been well explained and the figures are of high quality. So, all in all, I think this is a great manuscript with a very good fit for *Comms. Biol.*

Despite my enthusiasm for this work, I think the weakest part is the part on evolutionary adaptation. Although the right tools appear to have been used (e.g. PAML) and the reference genome appears to be of high quality, the authors have missed to give sufficient details about their population genomics analyses and associated results. My reason to believe that is as follows:

- A) It is unclear to me how the authors use the term 'substitution rate'. If this is supposed to be equivalent to mutation rate, the data shown in figure 2a are pretty high. Please provide a unit such as 'mutations per base per year'.
- B) As the expansion of SSRs is associated with the expansion of TEs, I suggest to calculate the expansion rate of these sequences using, for instance, Bayesian interference lineage diversification rate analysis (e.g. TESS for R). This will give evidence for the statements made in the paper about the expansion of any gene families under investigation (e.g. recent vs ancient TEs shown in figure 2c).
- C) Your phylogenetic results should be underpinned by divergence estimates in millions of years, for instance. This can be implemented in BEAST (e.g. Bouckaert et al. 2019). Dating will strengthen your results.
- D) It was unclear to me if genes involved in metabolic adaptation have also been expanded. If so, their divergence time should also be estimated using BEAST.
- E) If the expansion of TEs and some metabolic genes is the consequence of adaptive evolution, I wonder if dN/dS analyses would help to strengthen this argument.

More specific suggestions:

- 1) Why have the authors only focussed on the ATAC repeat to investigate the impact on gene expression? There are many more significant repeats as I have learned from the supplement. Is this the only repeat present in promoter regions? Please clarify.
- 2) Figure 5a should have a time line in 'millions of years' based on BEAST results.

- 3) If processes shown in figure 5b are really based on adaptive evolution, provide evidence (e.g. dN/dS, Fst, etc.)
- 4) The methods part called 'phylogenetic analysis' misses to explain the tools and approaches you have used for phylogenomics. Thus, please add details and provide additional data in the supplement.
- 5) Some figures lack statistics such as figure 1d.
- 6) Use 'single sequence repeats' in the title, not SSR.

Reviewer #2 (Remarks to the Author):

Review of SSRs drive genome plasticity and promote adaptive evolution in penaeid shrimp
This paper investigates the proportion of simple sequence repeats (SSRs) in the genomes of two penaeid decapods. Potential adaptive functions for the high proportions of SSRs in these genomes are investigated.

A large amount of work went into this paper and I do not see any major flaws in the methods or results. I think this is an important body of work, but I find it very long and hard to read. A large amount of the results section can be saved for the discussion or removed or simplified. There are some claims throughout the paper that need to be softened as well (See comments below).

Minor comments:

Suggest continuous line numbering to make it easier to refer to specific minor comments

L8: remove comma after chromosome-level

P3 L4: "junk DNA" in its original definition was never assumed to be purely non-functional. The original papers by Crick, Doolittle, etc. never assumed non-function. It would be more appropriate to say "non-coding DNA" than junk DNA.

P3 L26: why must they have function? Some DNA exists in our genomes simply because it can, and it has not become too costly to replicate yet at the organism level.

Much of the results section reads as Discussion. Stick to just the results in that section, and keep discussion separate.

P11 L25: change "around" to "adjacent" or "surrounding"

P15 L24: you say here that they belong to the same genus, but earlier in the paper you say you'll refer to them as their new genera names (or subgenera names)

P21 L22: this is a strong statement, and I suggest replacing with "and help regulate osmoregulation in penaeid shrimp". There are many factors involved in osmoregulation and adaptation to salinity.

P26 L1-20: I don't agree with much of this paragraph, or some of it needs re-wording. Much of it is reaching for an explanation, and comparing the SSR content here to mass extinctions is a bit of a reach in my opinion. Why are decapods assumed to have the highest ecological value among the crustaceans? SSRs would not necessarily be eliminated if they were "junk DNA" – as mentioned above, junk DNA was never assumed to be completely non-functional. Further, the opposing forces of genetic drift can allow for the expansion of SSRS and increase in genome size in the absence of natural selection in small effective population sizes (as may be expected after a mass extinction event).

P27 L2-5: No evidence is presented to support this claim

P27 L11-13: Reword sentences to “Why are dinucleotide SSRs specifically selected by penaeid shrimp” and “What other functions do SSRs serve in SSR-rich species?”

Conclusions and throughout – Penaeids are incorrectly referred to as “shrimp” when they are in fact “prawns” (Dendrobranchiata). Shrimps are caridean decapods. Similarly, the title should be changed to say “prawns” or “decapods”

Methods: was human blood used as a reference standard for flow cytometry? where was this obtained?

Reviewer #3 (Remarks to the Author):

Dear editor,

It's a pleasure to review the MS 6330 "SSRs drive genome plasticity and promote adaptive evolution in penaeid shrimp". This study reports the new genome assembly of *F. chinensis* and performed comparative genomics of two penaeid shrimp species. The authors found high abundance of SSRs in these species and that TEs are carriers for SSR. They showed that SSE insertion is related to chromosomal rearrangement. They performed extensive expression analysis to show that genes related to salinity adaptation have high SSR insertion in the regulatory regions. This study is important because many crustaceans are economically important but their genomes are large and complex. It's especially important to study the genomic underpinning of adaptive evolution in crustaceans because they are highly diverse and occur many marine and freshwater habitats.

While the author provided convincing evidence that SSRs are related to adaptive evolution in these shrimps, I cannot see why this effect is solely the result of SSRs but not also transposable elements (TEs), especially because the authors showed that TEs responsible for the propagation of SSRs. Theoretically, TE insertion could have similar effect in affecting chromosomal rearrangement, as well as gene regulation. I believe the authors need to provide some evidence why TE insertions are not causing these changes, and why they argue that, as the title says, SSR drive genome plasticity...". This has to be resolved for this MS to be accepted.

Minor comments are below.

Page 3 Line 22-24: "Even escape from MMR system, slippage mutations are unlikely to cause a major expansion of SSRs" Can you explain why?

Page 4 Line 20: How do you define "genome plasticity"?

Page 5 Line 3: remove "was"

Page 5 Line 12: Where are the genome assemblies deposited?

Page Line :

Page 8 Line 1-2: I'd argue rather that the highly plastic genome allowed these species to survive and adapt

Page 8 Line 7-11: Which other species that have undergone adaptive evolution are showing the same expanded gene families?

Page 8 Line 17: Looks like you used more than 2 SSR-rich species, you should at least provide a number here. Right now, it sounds like you compared only with *P. humanus* and *H. robusta*.

Page 8 Line 25-27: You'll need a robust analysis to make this statement. Right now, you can only say there's the types of expanded SSRs are lineage specific. The "Expansion of SSRs" regardless of type could be convergent. Who knows without a comparative analysis?

Page 9 Line 24: "different" instead of "difference"

Page 9 Line 25: Are the location of SSR identical between the two species. I think this should be presented and discussed

Page 9 Line 28-29: How do you attribute variation in species-specific expanded SSRs to adaptive evolution, rather than just divergence?

Page 11 Line 25: "surrounding" instead of "around"

Page 11 Line 26: "Similar to" instead of "Similar with"

Page 11 Line 28: "13.00% and 9.33%" are these the percentage of the genome or that of TEs?

Page 12 Line 28: " we doubted that these recent

28 TEs were not associated with these specifically expanded SSRs at the beginning." This sentence is very confusing. Please rewrite.

Page 18 Line 2: "we next studied WHETHER"

Page 18 Line 19: "shrimp specieS"

Page 19 Line 27: "would result IN variations"

Page 21 Line 18: Insertion in the intron is not supposed to affect the reading frame of the exon, unless the SSR contain sequences matching the splice donor or acceptor site. Can you back up your statement?

RESPONSE TO THE REVIEWER'S COMMENTS

Reviewers' comments:

Reviewer #1 (Remarks to the Author):

This manuscript explores the role of simple sequence repeats (SSRs) in the genomes of two penaeid shrimp species. The authors have done a very comprehensive analysis based on genome resequencing, de-novo sequencing, transcriptomics and metabolomics to shed light on how these repeats have evolved and how they contribute to the adaptation of shrimp species to their specific ecological niche. The main outcome of this study is that the repeats appear to drive adaptive evolution by increasing genome plasticity with consequences for the evolution of phenotypes that convey a fitness advantage under given environmental conditions. Overall, this appears to be a thoroughly conducted study encompassing very different approaches and methods to reveal the significance of repeats for the evolution of shrimp species. I was impressed by the amount of work the authors have put into this study, and their manuscript was carefully written up. Generally, results have been well explained and the figures are of high quality. So, all in all, I think this is a great manuscript with a very good fit for *Comms. Biol.*

Despite my enthusiasm for this work, I think the weakest part is the part on evolutionary adaptation. Although the right tools appear to have been used (e.g. PAML) and the reference genome appears to be of high quality, the authors have missed to give sufficient details about their population genomics analyses and associate results. My reason to believe that is as follows:

A) It is unclear to me how the authors use the term 'substitution rate'. If this is supposed to be equivalent to mutation rate, the data shown in figure 2a are pretty high. Please provide a unit such as 'mutations per base per year'.

Response:

The 'substitution rate' of TEs was calculated using RepeatMasker. It calculated the substitutions in matching region compared to the consensus. According to a previous study (Spears, 1992), the substitution rate of decapods is 2×10^{-9} substitutions per site per year. Thus, we re-estimated the approximate time of these TEs and revised the Figure 2A accordingly. As expected, the ancient TEs were estimated to have an expansion before the penaeid shrimp divergence, while recent TEs were estimated to expand after the penaeid shrimp divergence.

Reference: " Spears, T., Abele, L. G., & Kim, W. (1992). The Monophyly of Brachyuran Crabs - a Phylogenetic Study Based on 18s-Ribosomal-Rna. *Syst Biol*, 41(4), 446-461."

B) As the expansion of SSRs is associated with the expansion of TEs, I suggest to calculate the expansion rate of these sequences using, for instance, Bayesian

interference lineage diversification rate analysis (e.g. TESS for R). This will give evidence for the statements made in the paper about the expansion of any gene families under investigation (e.g. recent vs ancient TEs shown in figure 2c).

Response:

Thanks for reviewer's suggestion, we have tried to use TESS for the diversification rate analysis of TEs. However, we find it difficult to perform TESS analysis on TEs according to its manual instruction, and previous related studies are also limited. According to many previous studies (Xu, 2019; Hu, 2011; Slotte, 2013), the values of substitution rate per site per year was widely used to estimate the divergence time of TEs. Studies reported that decapods evolved at a rate of about 2×10^{-9} substitutions per site per year (Spears, 1992), thus, we used it to estimate the divergence time of these TEs. As expected, the ancient TEs were estimated to expand before the penaeid shrimp divergence, while recent TEs were estimated to expand after the penaeid shrimp divergence.

References:

Xu, P., Xu, J., Liu, G. J., Chen, L., Zhou, Z. X., Peng, W. Z., . . . Sun, X. W. (2019). The allotetraploid origin and asymmetrical genome evolution of the common carp *Cyprinus carpio*. *Nature Communications*, 10.

Hu, T. T., Pattyn, P., Bakker, E. G., Cao, J., Cheng, J. F., Clark, R. M., . . . Guo, Y. L. (2011). The *Arabidopsis lyrata* genome sequence and the basis of rapid genome size change. *Nature Genetics*, 43(5), 476.

Slotte, T., Hazzouri, K. M., Agren, J. A., Koenig, D., Maumus, F., Guo, Y. L., . . . Wright, S. I. (2013). The *Capsella rubella* genome and the genomic consequences of rapid mating system evolution. *Nature Genetics*, 45(7), 831-U165.

C) Your phylogenetic results should be underpinned by divergence estimates in millions of years, for instance. This can be implemented in BEAST (e.g. Bouckaert et al. 2019). Dating will strengthen your results.

Response:

Thanks for your suggestions. As we know, there are many methods to estimate the divergence times, so we used MCMCTREE of the PAML package to estimate the divergence time in the manuscript. Based on your suggestion, we redid the divergence time estimation of the phylogenetic tree using BEAST. The estimated time was very similar to the previous results. We revised the relevant results in the Results section (LINE 194) and described the BEAST methods in the Methods section.

D) It was unclear to me if genes involved in metabolic adaptation have also been expanded. If so, their divergence time should also be estimated using BEAST.

Response:

We did identify some expanded gene families related to metabolism of glycan, amino acid and lipid in two penaeid shrimp genomes, but these gene were not the same as identified in the differential expressed genes under low-salinity stress. According to your suggestion, we performed BEAST analyses on 36 expanded gene families, and estimated their divergence times. As expected, most of genes were expanded after the

divergence of two penaeid shrimp species, and some genes also divergent before the divergence of two penaeid shrimp species. We added the relevant results and discussions in the Results and Discussion sections, respectively. And a figure was added in the supplementary files (Supplementary Fig. 12)

E) If the expansion of TEs and some metabolic genes is the consequence of adaptive evolution, I wonder if dN/dS analyses would help to strengthen this argument.

Response:

Thank you for your suggestion. We performed positive selection analyses on the orthologous genes of penaeid shrimp species, and identified 148 positively selected genes. These genes were also enriched in the pathways related to amino acid metabolism, but are not the same as the expanded metabolic genes. We have added the relevant results in the Result sections.

More specific suggestions:

1) Why have the authors only focussed on the ATAC repeat to investigate the impact on gene expression? There are many more significant repeats as I have learned from the supplement. Is this the only repeat present in promoter regions? Please clarify.

Response:

In order to investigate the association between SSRs and gene expression regulation, the ATAC-seq is one of the best choice to be utilized, because it can widely identify the potential regulatory regions in the genome. The enrichment of SSRs in differential peaks indicating that these SSRs might play important roles in regulating gene expression. Whereas, RNA-seq and many other methods will not find these relationships. According to your suggestion, we performed additional analyses on other repeats (mainly TEs) in ATAC peaks. Similar to that of SSRs, the content of TEs in the differential peaks increased as the significance increased, but unchanged until p value $< 1.00E-5$. Thus, TEs and SSRs may both play roles in regulating gene expression. However, gene structure analysis of *glyA* indicated that TEs did not show obvious correlation with the two differential peaks. Instead, TEs are generally distributed around SSRs. We added the relevant result in the Results section, and provided additional figures in the supplementary files (Supplementary Fig. 29).

2) Figure 5a should have a time line in 'millions of years' based on BEAST results.

Response:

We utilized BEAST to estimate the time line in phylogenetic analysis, and we provided the geologic time of Mesozoic in Figure 5a.

3) If processes shown in figure 5b are really based on adaptive evolution, provide evidence (e.g. dN/dS, Fst, etc.)

Response:

Figure 5B is a pattern diagram to summarize the results of this study. We found that the expansion of SSRs is associated with the genome rearrangement and it is also related to the regulation of gene expression during low-salinity adaptation of shrimps.

So we suggested that SSRs may contribute to the adaptive evolution of penaeid shrimp. In order to avoid misleading readers, we deleted this figure.

4) The methods part called 'phylogenetic analysis' misses to explain the tools and approaches you have used for phylogenomics. Thus, please add details and provide additional data in the supplement.

Response:

We supplemented the methods of phylogenomics, and added the detail information on tree construction in Methods section according to your suggestions.

5) Some figures lack statistics such as figure 1d.

Response:

We added the statistic results in figure 1d, 1c, 2b, and the relevant figures in supplementary materials.

6) Use 'single sequence repeats' in the title, not SSR.

Response:

Thank you for your suggestion. We revised the title as you suggested.

Reviewer #2 (Remarks to the Author):

Review of SSRs drive genome plasticity and promote adaptive evolution in penaeid shrimp

This paper investigates the proportion of simple sequence repeats (SSRs) in the genomes of two penaeid decapods. Potential adaptive functions for the high proportions of SSRs in these genomes are investigated.

A large amount of work went into this paper and I do not see any major flaws in the methods or results. I think this is an important body of work, but I find it very long and hard to read. A large amount of the results section can be saved for the discussion or removed or simplified. There are some claims throughout the paper that need to be softened as well (See comments below).

Response:

Thanks for reviewer's suggestion. We have shortened the length of the manuscript to try to make it more concise. Some inappropriate claims were also removed from the Results and Discussion section. The major flaws of this work was indicated in the last paragraph of the Introduction section.

Minor comments:

Suggest continuous line numbering to make it easier to refer to specific minor comments

Response:

We used continuous line numbering now.

L8: remove comma after chromosome-level

Response:

We revised it according to your suggestion.

P3 L4: “junk DNA” in its original definition was never assumed to be purely non-functional. The original papers by Crick, Doolittle, etc. never assumed non-function. It would be more appropriate to say “non-coding DNA” than junk DNA.

Response:

We revised it according to your suggestion.

P3 L26: why must they have function? Some DNA exists in our genomes simply because it can, and it has not become too costly to replicate yet at the organism level.

Response:

We revised this sentence as follows:

"Generally, SSRs were considered to have no functions in the genome, whereas, many studies support an evolutionary role for some SSRs as important sources of adaptive genetic variation."

Much of the results section reads as Discussion. Stick to just the results in that section, and keep discussion separate.

Response:

We have deleted or moved much of discussions from Result section.

P11 L25: change “around” to “adjacent” or “surrounding”

Response:

We revised it according to your suggestion.

P15 L24: you say here that they belong to the same genus, but earlier in the paper you say you’ll refer to them as their new genera names (or subgenera names)

Response:

We changed this sentence to " these genomes showed a poor synteny even though they had close phylogenetic relationship".

P21 L22: this is a strong statement, and I suggest replacing with “and help regulate osmoregulation in penaeid shrimp”. There are many factors involved in osmoregulation and adaptation to salinity.

Response:

Thank you for your suggestion. We revised it according to your suggestion.

P26 L1-20: I don’t agree with much of this paragraph, or some of it needs re-wording. Much of it is reaching for an explanation, and comparing the SSR content here to mass extinctions is a bit of a reach in my opinion. Why are decapods assumed to have the highest ecological value among the crustaceans? SSRs would not necessarily be

eliminated if they were “junk DNA” – as mentioned above, junk DNA was never assumed to be completely non-functional. Further, the opposing forces of genetic drift can allow for the expansion of SSRS and increase in genome size in the absence of natural selection in small effective population sizes (as may be expected after a mass extinction event).

Response:

Thank you for your suggestion. We deleted some strong statements in this paragraph, and re-wording some sentences according to your suggestion. Here, we only made a statement of " During these two stages, an extreme expansion and species-specific expansion of SSRs were identified in penaeid shrimp genomes, suggesting the potential relationship between SSR expansion and shrimp adaptive evolution."

P27 L2-5: No evidence is presented to support this claim

Response:

We deleted this sentence.

P27 L11-13: Reword sentences to “Why are dinucleotide SSRs specifically selected by penaeid shrimp” and “What other functions do SSRs serve in SSR-rich species?”
Conclusions and throughout – Penaeids are incorrectly referred to as “shrimp” when they are in fact “prawns” (Dendrobranchiata). Shrimps are caridean decapods. Similarly, the title should be changed to say “prawns” or “decapods”

Response:

Thanks for your suggestions. We revised these sentences according to your suggestion. Prawns and shrimp are often confused. In the UK, Australia, New Zealand and Ireland, “prawn” is the general term used to describe both true prawns and shrimp. In North America, the term “shrimp” is used much more frequently, while the word “prawn” is most often used to describe larger species or those fished from fresh water. Since “shrimp” are widely used in other published papers and our previous works to represent Penaeidae species, thus, in order to avoid confusing, it was still used in this manuscript follow its previous usages.

Methods: was human blood used as a reference standard for flow cytometry? where was this obtained?

Response:

We used the human lymphocytes in the CYTO-COMP Cell Kit (No.CO6607023) following the instrument. In fact, we had also tried the mouse (*Mus musculus*, genome size of ~2.50 Gb) blood cells as internal standard for DNA content measures. In order not to cause misunderstanding, we use mouse blood cells as reference standard in the revised version.

Reviewer #3 (Remarks to the Author):

Dear editor,

It's a pleasure to review the MS 6330 "SSRs drive genome plasticity and promote adaptive evolution in penaeid shrimp". This study reports the new genome assembly of *F. chinensis* and performed comparative genomics of two penaeid shrimp species. The authors found high abundance of SSRs in these species and that TEs are carriers for SSR. They showed that SSE insertion is related to chromosomal rearrangement. They performed extensive expression analysis to show that genes related to salinity adaptation have high SSR insertion in the regulatory regions. This study is important because many crustaceans are economically important but their genomes are large and complex. It's especially important to study the genomic underpinning of adaptive evolution in crustaceans because they are highly diverse and occur many marine and freshwater habitats.

While the author provided convincing evidence that SSRs are related to adaptive evolution in these shrimps, I cannot see why this effect is solely the result of SSRs but not also transposable elements (TEs), especially because the authors showed that TEs responsible for the propagation of SSRs. Theoretically, TE insertion could have similar effect in affecting chromosomal rearrangement, as well as gene regulation. I believe the authors need to provide some evidence why TE insertions are not causing these changes, and why they argue that, as the title says, SSR drive genome plasticity...". This has to be resolved for this MS to be accepted.

Response:

Thanks for reviewer's valuable suggestions. The functions of TEs are eliminated in this work. Thus, we performed a thorough analysis on the functions of TEs according to the analyses of SSRs. Firstly, we tested the association of TEs and chromosomal rearrangement. As expected, in the plot of TE content and rearrangement sites along the genome (Fig. 3D), TEs displayed similar distributions with that of SSRs, but the sharp peaks of high TE content were identified adjacent to the peaks of SSRs generally. Secondly, we calculated the TE content of the ATAC peaks to find the relationship between TEs and gene regulation. Similar to that of SSRs, the content of TEs in the differential peaks increased as the significance increased, but unchanged until p value $< 1.00E-5$ (Supplementary Fig. 29). Besides, in the plot of gene structure (Fig. 4F), TEs were generally identified adjacent to SSRs. Therefore, both results supported our conclusion that TEs may serve as carriers for SSR expansion. As the important component of the genome, TEs are also the important factors resulting for genomic plasticity. Thus, we added a paragraph to discuss the potential functions of TEs and SSRs in the Discussion section.

Minor comments are below.

Page 3 Line 22-24: "Even escape from MMR system, slippage mutations are unlikely to cause a major expansion of SSRs" Can you explain why?

Response:

SSRs account for more than 23% of the penaeid shrimp genome. If slippage mutations result for the mutation of 23% of the genome, their DNA replication system will not stable. However, this sentence may be in-appropriate to be there, thus we deleted it.

Page 4 Line 20: How do you define "genome plasticity"?

Response:

We made a definition of "genome plasticity" ahead of this paragraph.

Page 5 Line 3: remove "was"

Response:

We revised it according to your suggestion.

Page 5 Line 12: Where are the genome assemblies deposited?

Response:

We made a statement of Data availability at the end of the manuscript. Genome sequence data that support the findings of this study have been deposited in NCBI GenBank with the accession number of JABKCB000000000 and BioProject accession codes of PRJNA627295. The *F. chinensis* genome and the annotation information can also be downloaded from the Shrimp Genome Database (http://www.genedatabase.cn/fch_genome.html).

Page Line :

Page 8 Line 1-2: I'd argue rather that the highly plastic genome allowed these species to survive and adapt

Response:

We agree with you. This sentence reads as discussion, thus, we deleted this sentence according to the suggestions of Reviewer 2.

Page 8 Line 7-11: Which other species that have undergone adaptive evolution are showing the same expanded gene families?

Response:

These gene families were identified specifically expanded in penaeid shrimp genomes, which was different from other species. Different species may develop different mechanism for adaptive evolution.

Page 8 Line 17: Looks like you used more than 2 SSR-rich species, you should at least provide a number here. Right now, it sounds like you compared only with *P. humanus* and *H. robusta*.

Response:

We have calculated the SSR content among 36 eukaryotic species, and among them, *P. humanus* and *H. robusta* are two species with the highest amount of SSRs. Thus, we compared them with penaeid shrimp species in detail. We have added a new table (Supplementary table 11) to show the statistic results of these species. Beside *P. humanus* and *H. robusta*, we also compared the two shrimp species with that of *E. sinensis*, which is another SSR-rich decapod (Fig. 1F).

Page 8 Line 25-27: You'll need a robust analysis to make this statement. Right now,

you can only say there's the types of expanded SSRs are lineage specific. The "Expansion of SSRs" regardless of type could be convergent. Who knows without a comparative analysis?

Response:

We revised this statement according to your suggestion. It was revised as follows:

" Thus, the expansion of some specific types of SSRs was an independent lineage-specific event rather than a convergent evolution event in SSR-rich species."

Page 9 Line 24: "different" instead of "difference"

Response:

We revised it according to your suggestion.

Page 9 Line 25: Are the location of SSR identical between the two species. I think this should be presented and discussed

Response:

Thank you for your suggestion. The location of SSRs showed significant difference between the two species. We have provided the relevant results in the Result section, and also discussed it in detail. As we shown in the following results of genome synteny and gene structure comparison, we found that the two genomes displayed poor synteny, even in high consensus gene clusters (Hox gene cluster). Thus, the location of SSRs were also significantly altered between two genomes. Then, we compared the location of SSRs with the orthologous genes of two species (Supplementary Fig. 25), we found that the composition of the SSRs located within introns or UTRs showed significant differences in most orthologous genes, even though exons displayed high synteny between the two shrimp species. A high number of new SSRs have been inserted or deleted in these genes, including some extraordinarily long SSRs, e.g., (AT)₆₇₆ in the intron 2 of GNMT and (AT)₆₃₇ in the intron 2 of PSPH. In addition, some SSRs were significantly elongated in one shrimp species compared to the other despite being located at the same position in an orthologous gene, e.g., (AG)_n in the intron 5 of DMGDH had the motif repeat numbers of 53 and 421 in *F. chinensis* and *L. vannamei*, respectively. Even for SSR-free genes (e.g., Klf1), the SSRs located in the UTRs also showed some differences.

Page 9 Line 28-29: How do you attribute variation in species-specific expanded SSRs to adaptive evolution, rather than just divergence?

Response:

Thank you for your suggestion, we revised this sentence as follows:

"However, some variations, especially species-specific expanded SSRs, were also identified between the two genomes, which occurred after their divergence."

Page 11 Line 25: "surrounding" instead of "around"

Response:

We revised it according to your suggestion.

Page 11 Line 26: "Similar to" instead of "Similar with"

Response:

We revised it according to your suggestion.

Page 11 Line 28: "13.00% and 9.33%" are these the percentage of the genome or that of TEs?

Response:

"13.00% and 9.33%" are the percentage of the genome. We revised it as "13.00% and 9.33% of the genomes for *F. chinensis* and *L. vannamei*, respectively".

Page 12 Line 28: " we doubted that these recent TEs were not associated with these specifically expanded SSRs at the beginning." This sentence is very confusing. Please rewrite.

Response:

We deleted this sentence.

Page 18 Line 2: "we next studied WHETHER"

Response:

We revised it according to your suggestion.

Page 18 Line 19: "shrimp species"

Response:

We revised it according to your suggestion.

Page 19 Line 27: "would result IN variations"

Response:

We revised it according to your suggestion.

Page 21 Line 18: Insertion in the intron is not supposed to affect the reading frame of the exon, unless the SSR contain sequences matching the splice donor or acceptor site. Can you back up your statement?

Response:

We revised this statement as " an SSR insertion ((CCTTT)₇₃) was also identified near the exon 3 in *F. chinensis*, which might potentially affect its transcription. "

REVIEWERS' COMMENTS:

Reviewer #1 (Remarks to the Author):

Many thanks for addressing all of my comments. This is really a very interesting paper.
Congrats!

Reviewer #2 (Remarks to the Author):

I believe the authors have adequately revised the text of this manuscript and it is suitable for publication. This will serve as a good resource for crustacean genomics, which are currently lacking.

Some minor typos need fixing and I suggest one final proof-read of the manuscript before final submission:

L35: should read "are generally considered to have no function"

L87 (of the track changes .docx file): suggest changing this sentence to "Thus, aside from slippage mutations, additional mechanisms underlying SSR expansion may exist in SSR-rich species" (check for English grammar). Further, I suggest not deleting the sentence before this, which is currently removed.

L858 (of the track changes .docx file): "Relaxed" is spelled incorrectly

RESPONSE TO THE REVIEWER'S COMMENTS

REVIEWERS' COMMENTS:

Reviewer #1 (Remarks to the Author):

Many thanks for addressing all of my comments. This is really a very interesting paper. Congrats!

Response:

Thank you very much for reviewing our manuscript and providing much valuable suggestions.

Reviewer #2 (Remarks to the Author):

I believe the authors have adequately revised the text of this manuscript and it is suitable for publication. This will save as a good resource for crustacean genomics, which are currently lacking.

Response:

Thank you very much for reviewing our manuscript and providing much valuable suggestions.

Some minor typos need fixing and I suggest one final proof-read of the manuscript before final submission:

L35: should read "are generally considered to have no function"

Response:

We revised it according to your suggestion.

L87 (of the track changes .docx file): suggest changing this sentence to "Thus, aside from slippage mutations, additional mechanisms underlying SSR expansion may exist in SSR-rich species" (check for English grammar). Further, I suggest not deleting the sentence before this, which is currently removed.

Response:

We revised it according to your suggestion.

L858 (of the track changes .docx file): "Relaxed" is spelled incorrectly

Response:

We revised it according to your suggestion.